# The ortholog of human ssDNA-binding protein SSBP3 influences neurodevelopment and autism-like behaviors in *Drosophila melanogaster*

**Safa Salim**[1], **Sadam Hussain**[1], **Ayesha Banu**[1], **Swetha B. M. Gowda**[1], **Foysal Ahammad**[1], **Amira Alwa**[1], **Mujaheed Pasha**[2], **Farhan Mohammad**[1]*

**1** Division of Biological and Biomedical Sciences (BBS), College of Health & Life Sciences (CHLS), Hamad Bin Khalifa University (HBKU), Doha, Qatar, **2** HBKU Core Labs, Hamad Bin Khalifa University (HBKU): Doha, Qatar

* mohammadfarhan@hbku.edu.qa, farhan8igib@gmail.com

**Data Availability Statement:** All RNA-seq files are available from the NCBI GEO database (accession number(s) GSE220311) and all other relevant data

## Abstract

1p32.3 microdeletion/duplication is implicated in many neurodevelopmental disorders-like phenotypes such as developmental delay, intellectual disability, autism, macro/microcephaly, and dysmorphic features. The 1p32.3 chromosomal region harbors several genes critical for development; however, their validation and characterization remain inadequate. One such gene is the single-stranded DNA-binding protein 3 (*SSBP3*) and its *Drosophila melanogaster* ortholog is called sequence-specific single-stranded DNA-binding protein (*Ssdp*). Here, we investigated consequences of *Ssdp* manipulations on neurodevelopment, gene expression, physiological function, and autism-associated behaviors using *Drosophila* models. We found that SSBP3 and Ssdp are expressed in excitatory neurons in the brain. *Ssdp* overexpression caused morphological alterations in *Drosophila* wing, mechanosensory bristles, and head. *Ssdp* manipulations also affected the neuropil brain volume and glial cell number in larvae and adult flies. Moreover, *Ssdp* overexpression led to differential changes in synaptic density in specific brain regions. We observed decreased levels of armadillo in the heads of *Ssdp* overexpressing flies, as well as a decrease in armadillo and wingless expression in the larval wing discs, implicating the involvement of the canonical Wnt signaling pathway in Ssdp functionality. RNA sequencing revealed perturbation of oxidative stress-related pathways in heads of *Ssdp* overexpressing flies. Furthermore, *Ssdp* overexpressing brains showed enhanced reactive oxygen species (ROS), altered neuronal mitochondrial morphology, and up-regulated fission and fusion genes. Flies with elevated levels of *Ssdp* exhibited heightened anxiety-like behavior, altered decisiveness, defective sensory perception and habituation, abnormal social interaction, and feeding defects, which were phenocopied in the pan-neuronal *Ssdp* knockdown flies, suggesting that *Ssdp* is dosage sensitive. Partial rescue of behavioral defects was observed upon normalization of *Ssdp* levels. Notably, *Ssdp* knockdown exclusively in adult flies did not produce behavioral and functional defects. Finally, we show that optogenetic manipulation of Ssdp-expressing neurons altered autism-associated behaviors. Collectively, our findings provide evidence that *Ssdp*,

are within the paper and its Supporting Information files.

**Funding:** This study was supported partly by grants from Qatar National Research Fund (QNRF), to F.M.: UREP28-269-1-051, NPRP13S-0121-200130 and NPRP14S-0319-210075. F.M., S.S., S. H., S.B.M.G., A.B., A.S., and F.A. were further supported by the College of Health and Life Sciences (CHLS), Hamad Bin Khalifa University (HBKU), Qatar Foundation. The funders had no role in study design, data collection and analysis, decision to publish, or preparation of the manuscript.

**Competing interests:** The authors have declared that no competing interests exist.

**Abbreviations:** ASD, autism spectrum disorder; CNS, central nervous system; CNV, copy number variation; DART, Drosophila arousal tracking; DEG, differentially expressed gene; FlyPAD, fly proboscis and activity detectors; fps, frames per second; ID, intellectual disability; MIP, maximum intensity projection; MPA, minor physical anomalies; NDD, neurodevelopmental disorder; qRT-PCR, quantitative real-time PCR; RLE, relative log expression; RNA-seq, RNA sequencing; ROS, reactive oxygen species; SEM, scanning electron microscope; SEZ, subesophageal zone; SLP, superior lateral protocerebrum; SSBP3, single-stranded DNA binding protein 3; Ssdp, sequence-specific single-stranded DNA-binding protein.

a dosage-sensitive gene in the 1p32.3 chromosomal region, is associated with various anatomical, physiological, and behavioral defects, which may be relevant to neurodevelopmental disorders like autism. Our study proposes *SSBP3* as a critical gene in the 1p32.3 microdeletion/duplication genomic region and sheds light on the functional role of *Ssdp* in neurodevelopmental processes in *Drosophila*.

## Introduction

Autism spectrum disorders (ASD) are a group of heterogeneous neurodevelopmental disorders (NDDs) caused by multiple genetic/genomic dysfunctions, including chromosomal rearrangements, microdeletions, copy number variations (CNVs), and point mutations [1]. Autism is reported to be 4 times more prevalent in males than in females [2–4]. Around 15% to 20% of patients with NDDs, such as ASD, harbor chromosomal microdeletions [5–7]. As many as 23 patients with 1p32.3 microdeletion and 34 patients with 1p32.3 duplication are listed in the DECIPHER database [8], with many reported to have developmental delay, macrocephaly, intellectual disability (ID), and autism [9]. Notably, DECIPHER database has a record of 98 autistic patients who have micro/macrocephaly and many of these patients have 1p32.3 microdeletion or duplication [8]. Changes in the dosage of some genes by deletion or duplication can cause NDDs including ASD and ID [10]. Further, dosage sensitivity has been proposed to be a predominant causative factor underlying CNV pathogenicity [11]. Although the link between gene dosage sensitivity and diseases is well established, the mechanism of the pathogenicity remains unclear.

In humans, single-stranded DNA-binding protein 3 (*SSBP3*) is located in the 1p32.3 chromosomal region. A recent genome-wide assessment of the population frequency of deletion and duplication of CNVs determined *SSBP3* to be one of the genes showing significant evidence of both haploinsufficiency and triplosensitivity [12]. Suggesting that a microdeletion of *SSBP3* or a duplication that contributes an additional copy of the entire gene may alter the dosage of SSBP3 and produce deleterious phenotypes.

*SSBP3* is an evolutionarily conserved gene, having an ortholog in *Drosophila melanogaster* [13]. While humans possess 4 *SSBP3* homologs (*SSBP1*, *SSBP2*, *SSBP3*, and *SSBP4*), the fly genome has only 1; sequence-specific single-stranded DNA-binding protein (*Ssdp*). The protein sequences of SSBP3 and Ssdp contain highly conserved domains, including a lissencephaly type-1-like homology (LisH) motif and a proline-rich domain [13,14]. These regions in SSBP3 play a significant role in head development [14,15]. Furthermore, Ssdp is a part of the Wnt enhanceosome, which mediates the transcription switches of the Wnt/β-catenin signaling [16].

Genetic tractability and rich behavioral repertoire make *Drosophila* an excellent model for delineating genetic mechanisms of genes associated with psychiatric and cognition disorders, including ASD [17–24]. To determine the role of CNVs of *SSBP3* in neurodevelopment and autism-associated behaviors, we used *Ssdp* overexpression and pan-neuronal knockdown strategy in male *Drosophila* fruit flies. We show that *Ssdp* overexpression modifies the morphology of wing, bristles, and head, and alterations in *Ssdp* levels affect brain development and glial number. Synaptic density was altered differentially in different brain regions upon *Ssdp* overexpression. *Ssdp* overexpression affects brain and wing development potentially via its role in canonical Wnt signaling. Using RNA sequencing, we found that *Ssdp* overexpression causes differential regulation of numerous genes and affects molecular pathways related to

oxidative stress. *Ssdp* overexpression led to increased reactive oxygen species (ROS) and defective mitochondrial morphology and function. *Ssdp* overexpression also produced autism-associated behavioral deficits, and most of these features were recapitulated by *Ssdp* knockdown. Many of these behavioral defects were rescued upon normalization of *Ssdp* levels. Additionally, behavioral and functional defects were not observed upon temporal *Ssdp* knockdown in adult flies. We further show that optogenetically manipulating neuronal activity of Ssdp-expressing neurons in *Drosophila* altered various ASD-associated behavioral phenotypes. Overall, our data suggest that the *Ssdp* is a dosage-sensitive gene and the defects observed may in part be due to altered gliogenesis and Wnt signaling pathway. Taken together, our study proposes that *SSBP3* dosage alterations underlie the various ASD and ID phenotypes associated with the 1p32.3 microdeletion/duplication region.

## Results

### SSBP3/Ssdp expresses in excitatory neurons in humans and *Drosophila* brains

In humans, the 1p32.3 chromosomal region harbors multiple genes (S1 Table), including autism candidate genes; Glis Family Zinc Finger 1 (*GLIS1*) [25,26], Carnitine Palmitoyl Transferase 2 (*CPT2*) [27], Sterol Carrier Protein 2 (*SCP2*) [27], Ubiquitin Specific Peptidase 24 (*USP24*) [28], *SSBP3* and *SSBP3* antisense (*SSBP3-AS1*) [29] (https://omim.org/entry/607390) (Fig 1A). Out of these 6 autism-related genes, only 3 (*CPT2*, *SCP2* and *SSBP3*) have strong orthologs in flies (DIOPT Score >0.5, S1 Table), with the highest orthology for *SSBP3* (DIOPT Score = 0.93) (Fig 1B and S1 Table). Interestingly, Johnson and colleagues performed a systems-level analysis of genome-wide gene expression data to identify gene-regulatory networks associated with healthy cognitive abilities and neurodevelopmental disorders. One of the identified networks, M3, consists of 150 genes that also includes *SSBP3*, and M3 was found to be enriched for mutations in patients with ID and epileptic encephalopathy. Their results suggest a convergent gene-regulatory network influencing cognition and neurodevelopmental disease [30]. Furthermore, *SSBP3* is one of the 321 candidate genes prioritized in a cross-disorder analysis of de novo mutations, showing significant genetic association with neurodevelopmental disorders such as ASD and ID [31]. Additionally, a recent GWAS study identified multiple allele variants that map the *SSBP3* gene in the 1p32.3 chromosomal region and are associated with altered brain morphology, cortical surface area, and subcortical volume with a high statistical significance ($p$-value $< 10^{-21}$) [32]. Also, transgenic tools (RNAi and overexpressing alleles) are only available for the *SSBP3* ortholog in *Drosophila* (S1 Table). Hence, we selected *SSBP3* for downstream analysis (Fig 1B). Human-SSBP3 (Hs-SSBP3) protein is highly conserved and similar to *Drosophila* Ssdp (57% similar and 51% identical), consisting of highly conserved LisH (approximately 97% identical) and proline-rich (approximately 54% identical) domains (Fig 1C).

We analyzed SSBP3 expression in human brain using the human single-cell RNA-seq data (M1–10× GENOMICS (2020)) [33] available at Allen brain map (https://celltypes.brain-map.org/rnaseq). We found that SSBP3 colocalizes with markers of excitatory glutamatergic neurons; SLC17A7, CUX2, and RORB (Fig 2A and 2B) and with oligodendrocytes (Fig 2B); however, the expression level of SSBP3 was low compared to the markers (Fig 2B). Next, we sought to characterize Ssdp[2082-G4]-Gal4 and examined *Ssdp* expression pattern in the fly brain by driving 6xUAS-GFP [34]. *Ssdp* was expressed strongly in the superior lateral protocerebrum (SLP) and the subesophageal zone (SEZ) in the brain (Fig 2C). Using SCOPE (http://scope.aertslab.org) [35,36], we further determined that *Ssdp* is expressed in vGlut- (ortholog of

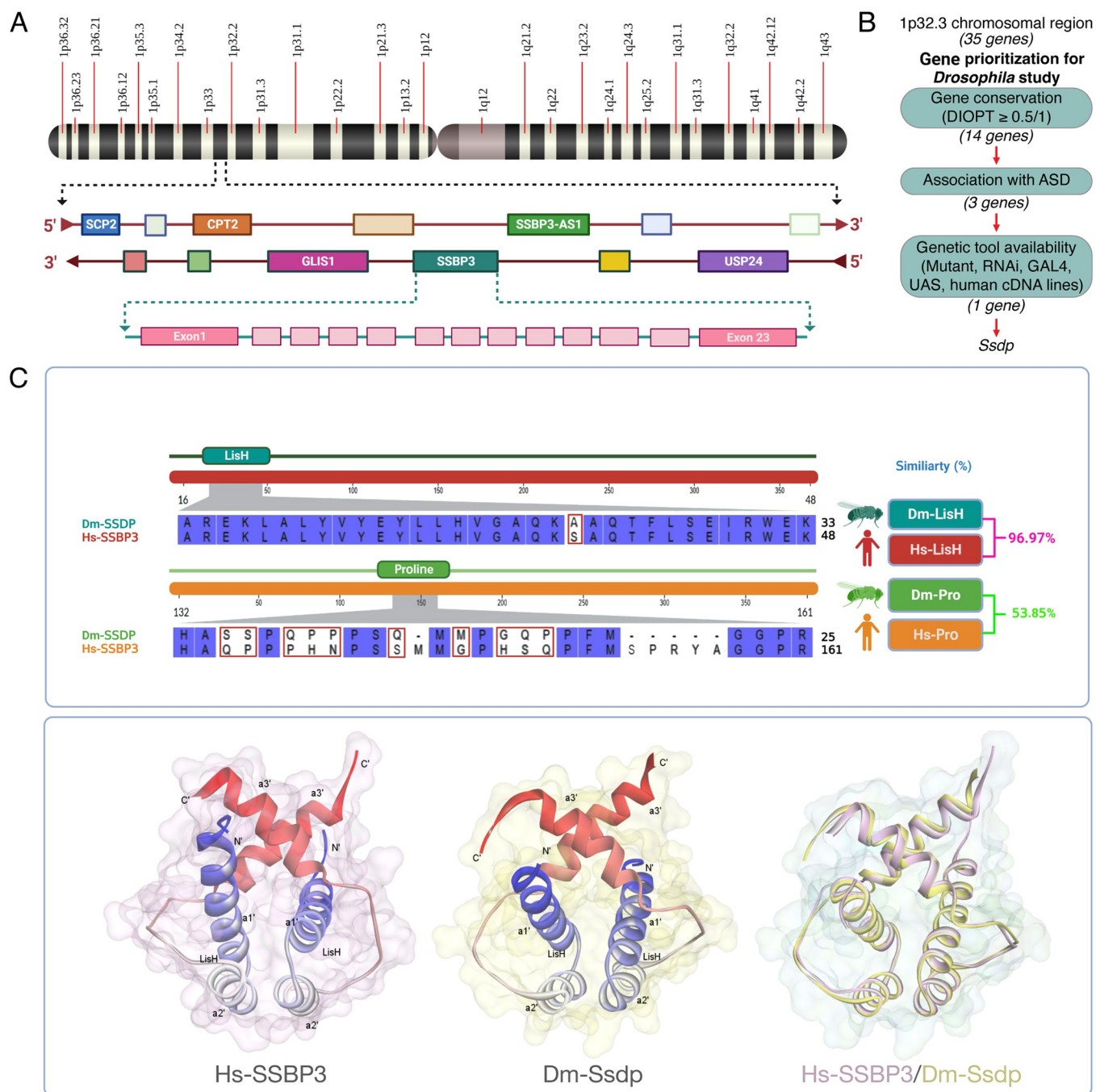

**Fig 1. *Ssdp* gene prioritization and protein homology.** (A) Human *SSBP3* gene is located in the 1p32.3 chromosomal region and has 23 exons. Created with BioRender.com. (B) Criteria to prioritize *Ssdp* as an ASD candidate gene for the study. (C) Sequence alignment of the LisH and proline-rich domains of Dm-Ssdp and Hs-SSBP3 (top). Ribbon representations of Hs-SSBP3 (PDB ID: 6IWV), *Ssdp* (PDB ID: 6S9R), and their superimposition (bottom). Created with Biovia discovery studio visualizer. ASD, autism spectrum disorder; SSBP3, single-stranded DNA-binding protein 3.

human SLC17A7) and reversed polarity (repo)-positive cells, which is a marker of glial cells (Fig 2D).

Ssdp[2082-G4] is an insertion allele generated using mobilization of a transposable element [34] (S1 Fig). Although Ssdp[2082-G4] has been characterized for capturing *Ssdp* neuronal expression pattern in the brain, the impact of the insertion on *Ssdp* loci has not been explored

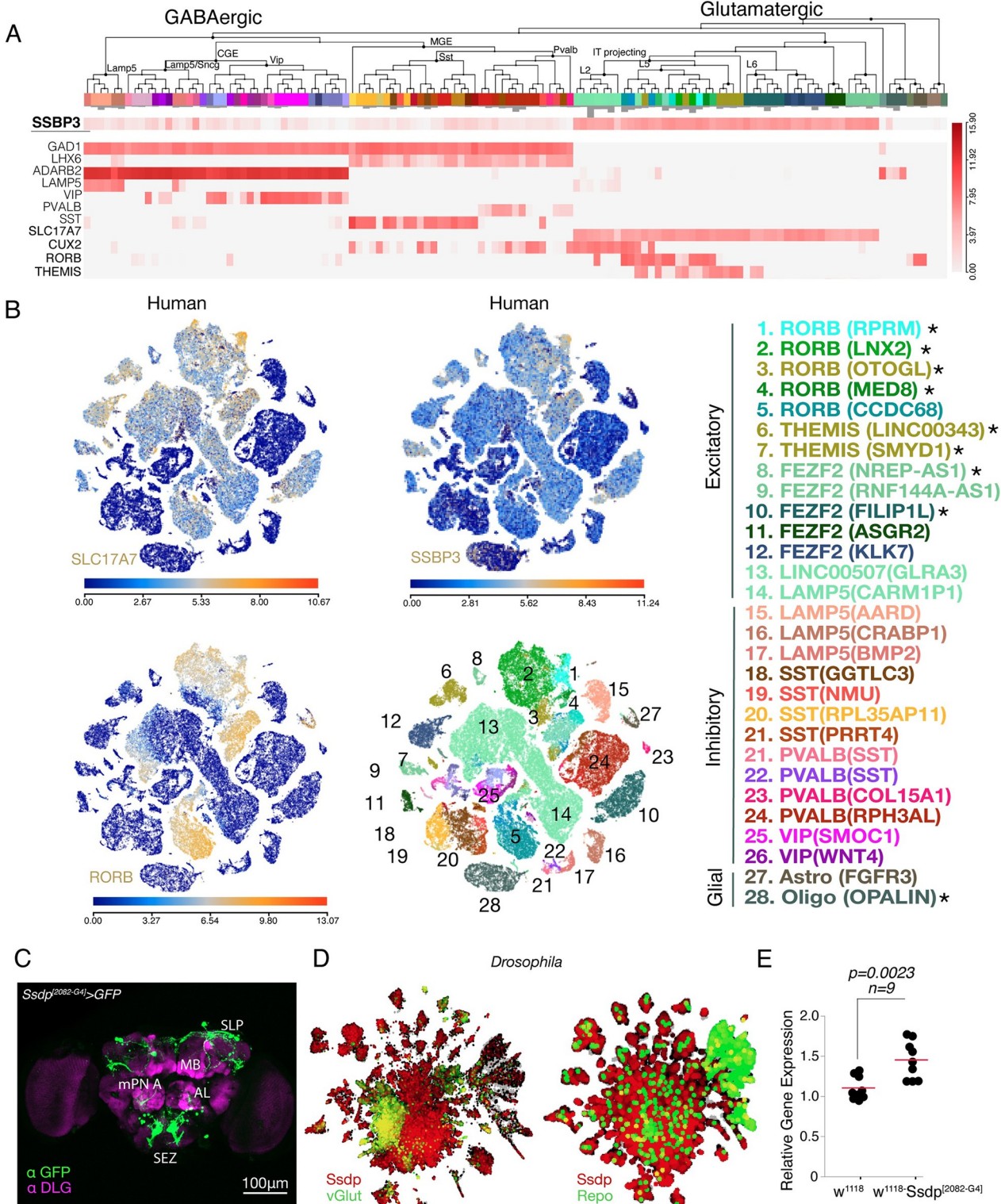

**Fig 2. SSBP3/Ssdp is expressed in the excitatory neurons in the human and adult fly brains.** (A) Heatmap depicting expression of SSBP3 in various cell types in the human brain using single-cell RNA-sequencing. (B) Scatter plots depicting the expression of SSBP3 in human brain cells that overlap with the cells expressing SLC17A7 and RORB. Heatmaps below scatterplots indicate expression level of indicated gene in each cell. In the bottom scatter plot, cell types are indicated by colors and numbers. Asterisks beside the names of cell types represent overlap of SSBP3 with excitatory and oligodendrocyte cell types in human brain. (C) Immunohistochemistry image showing Ssdp[2082-G4]>6x-UAS-GFP brain, stained with anti-DLG (magenta) and anti-GFP (green). Scale bar = 100 μm. (D) *Drosophila* single-cell transcriptomic data analysis reveals that Ssdp (red)

co-expresses with vGlut (green) and reversed polarity (Repo, green). The yellow color indicates colocalization. (E) Relative mRNA expression of *Ssdp* is increased in heterozygous Ssdp[2082-G4]/+ flies ($p = 0.0023$, $n = 9$). Each black dot is a data point representing an independent biological replicate with 20 individual fly heads per biological replicate. Total cDNA concentration was normalized to endogenous *Rpl32* expression. The horizontal red line represents the mean value of the scatter. *P*-values are calculated using the Student's *t* test. The raw data underlying panel 2E can be found in S1 Data file. SLC17A7, solute carrier family 17 member 7; RORB, RAR related orphan receptor B; SLP, superior lateral protocerebrum; MB, mushroom body; AL, antenna lobe; SEZ, suboesophageal zone; SSBP3, single-stranded DNA binding protein 3.

so far. Insertion of the transposable element or the genetic background may result in either loss-of-function or overexpression of the genes that are downstream of the targeted locus [37–40]. Moreover, within the intergenic region of *Ssdp* in the *Drosophila* genome, adjacent to the Ssdp-IT.Gal4 insertion, there exists a natural transposable element known as H.M.S. Beagle, the influence of which on the *Ssdp* transcript remains uncertain (S1 Fig). Nonetheless, the H. M.S. Beagle sequence includes a putative enhancer-like element that may induce transcription of nearby loci [41]. Since Ssdp[2082-G4] insertion is developmentally lethal in homozygous condition, we used heterozygous Ssdp[2082-G4] flies crossed with w[1118] in this study. To check the impact of the insertion element and H.M.S. Beagle on transcription of *Ssdp*, we performed qRT-PCR on the heads of 3- to 4-day-old F1 progeny of Ssdp[2082-G4] flies crossed with w[1118]. We found up-regulation of *Ssdp* mRNA in heterozygous Ssdp[2082-G4] flies in the w[1118] background, in comparison to w[1118] controls (Fig 2E).

## *Ssdp* overexpression leads to alterations in *Drosophila* wing, mechanosensory bristles, and head size

Minor physical anomalies (MPAs) are subtle morphological abnormalities of the face, head, and limbs, which are suggested to represent external markers of atypical brain development in ASD [42–44]. Studies have suggested a link between autistic traits, overall level of functioning, and MPAs [43,44]. Of note, craniofacial anomalies are a recurring feature of a subpopulation of ASD children with distinctive morphologies, including decreased facial midline height and long width of mouth, with ID and increased severity of ASD symptoms being comorbidities [45,46]. This comes as no surprise since both brain and face have common origins from neuroectodermal tissue and their development is closely coordinated due to their physical proximity and mutual molecular coordination [47]. Concomitantly, we asked whether overexpression of *Ssdp* produces any morphological abnormalities that phenocopy MPAs in ASD children. We performed scanning electron microscopy (SEM) with high magnification to detect structural changes in the wing, head, thorax, and eyes of 3- to 4-day-old Ssdp[2082-G4] adult flies compared to w[1118] controls. We observed a novel phenotype in the wings of Ssdp[2082-G4] flies, with enhanced and deeper indentations on the surface of the wing in comparison to controls (Fig 3A). An increase in the number of interocellar bristles was observed in Ssdp[2082-G4] flies compared to controls (Fig 3B). There was also loss of bristles on the surface of pedicel on the antenna of Ssdp[2082-G4] flies compared to controls (Fig 3E). Lastly, overall, the Ssdp[2082-G4] heads appeared larger than the heads of w[1118] controls (Fig 3E). No phenotypic differences were observed in the thorax, eyes, and proboscis of Ssdp[2082-G4] flies and controls (Fig 3C–3E, respectively). Our data suggest that *Ssdp* functions early in development and regulates the morphogenesis of wings, mechanosensory bristles, and head.

## *Ssdp* manipulations cause defects in brain volume, synaptic density, and glial cell number

Given the structural changes observed upon *Ssdp* overexpression in the head, we sought to determine the role of *Ssdp* in brain development, specifically in the regulation of structural

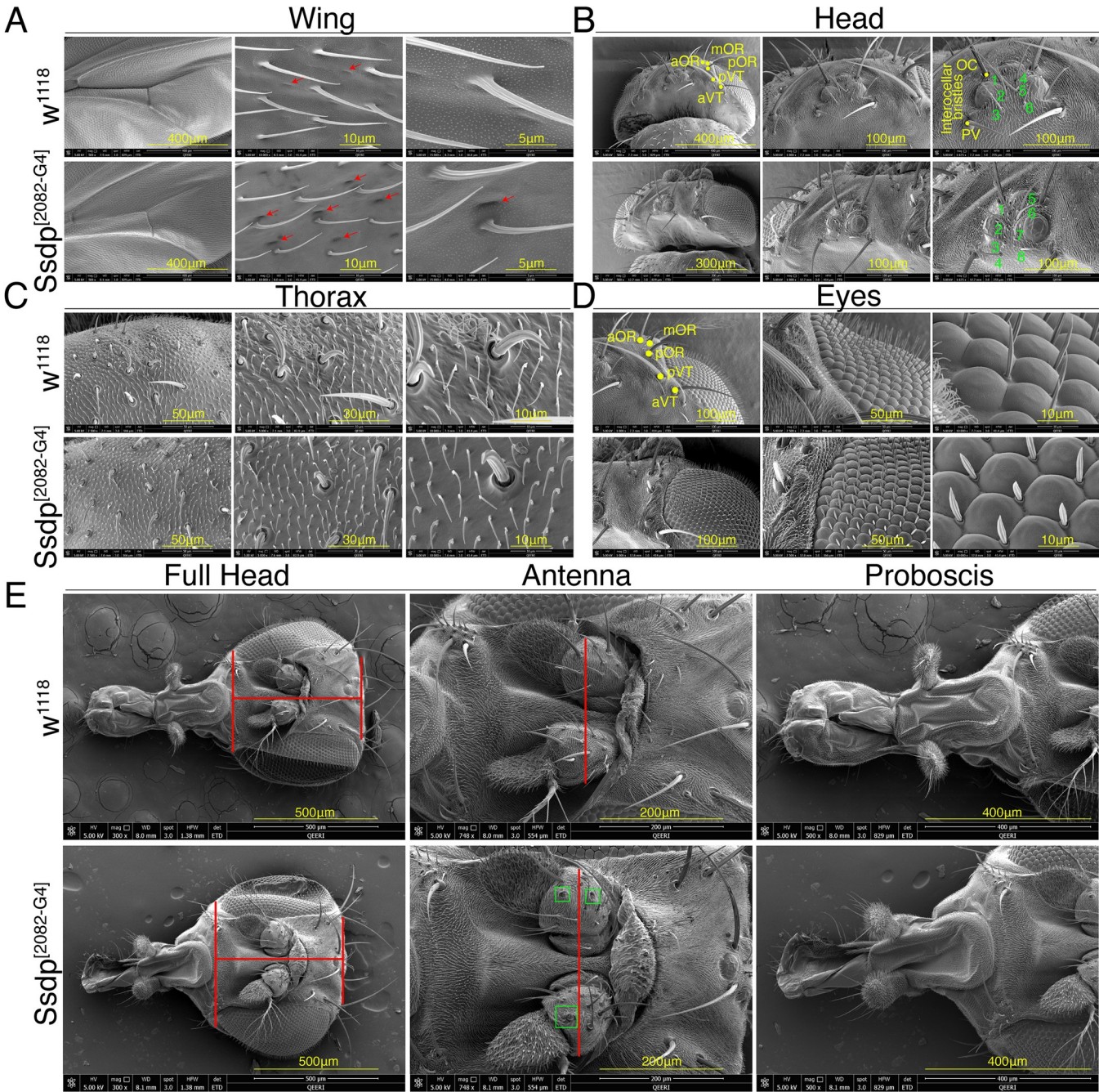

**Fig 3. *Ssdp* overexpression alters morphology of wing, mechanosensory bristles, and head.** High-resolution SEM images showing wing, head, thorax, and eyes of heterozygous Ssdp[2082-G4]/+ flies compared to w[1118] controls at high magnification. (A) Deeper indentations are observed on the surface of the wings of Ssdp[2082-G4]/+ flies compared to controls. Red arrows point towards the difference of phenotype. (B) The number of interocellar bristles is increased in Ssdp[2082-G4]/+ flies compared to controls. Numbers in green mark the increased number of interocellar bristles. (C) No phenotypic differences are observed in the bristles present on the thorax of Ssdp[2082-G4]/+ flies and controls. (D) No phenotypic differences are observed in the eyes of Ssdp[2082-G4]/+ flies and controls. (E) Overall, the head of Ssdp[2082-G4]/+ fly is larger than that of control. Some bristles on the surface of the pedicel on the antenna of Ssdp[2082-G4]/+ flies are lost compared to controls. No phenotypic differences are observed in proboscis of both Ssdp[2082-G4]/+ flies and controls. Red lines highlight increase in head size. Green squares depict loss of bristles on the pedicel. Scale bars are depicted at the bottom right of each image. aOR, anterior orbital bristles; mOR, middle orbital bristles; pOR, posterior orbital bristles; pVT, posterior verticel bristles; aVT, anterior vertical bristles; OC, ocellar bristles; PV, postvertical bristles; SEM, scanning electron microscope; Ssdp, sequence-specific single-stranded DNA-binding protein.

boundaries, cell types, and cell number in both Ssdp[2082-G4] and pan-neuronal *Ssdp* knock-down third instar larvae and 3- to 4-day-old adult flies. Interestingly, in both Ssdp[2082-G4] larvae and adult flies, we observed larger neuropil central nervous system (CNS) and brain volumes, respectively, labeled by anti-DLG antibody when compared to the w[1118] controls (Fig 4A and 4B and S1–S4 Movies). We further confirmed the increase in brain volume using

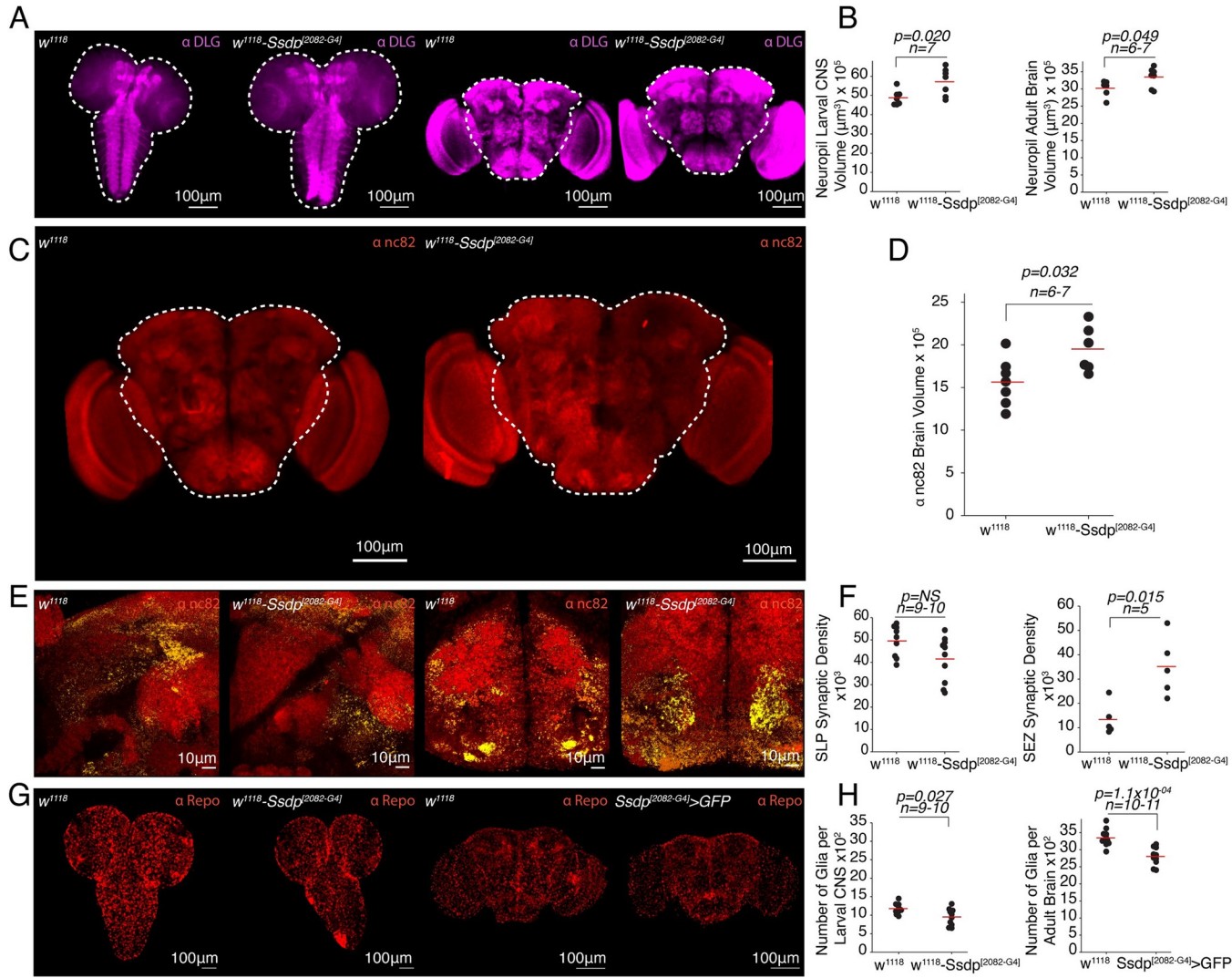

**Fig 4. *Ssdp* regulates neuropil development, synaptic density, and glial numbers.** (A) MIP images showing brain neuropils labeled by anti-DLG antibody in w[1118] and Ssdp[2082-G4]/+ larval CNS and adult brains. White dashed outlines depict CNS in larvae and central brain region in adults, which was used to analyze neuropil volume. Scale bar = 100 μm. (B) Neuropil brain volume is larger in both larval CNS ($p = 0.020$, $n = 7$) and adult brains (0.049, $n = 6–7$) of Ssdp[2082-G4]/+ flies compared to controls. (C) MIP images showing brain neuropils labeled by anti-nc82 antibody in w[1118] and Ssdp[2082-G4]/+ brains. White dashed outline depicts central brain region used to analyze brain volume. Scale bar = 100 μm. (D) Neuropil brain volume is larger in Ssdp[2082-G4]/+ flies compared to controls ($p = 0.032$, $n = 6–7$). (E) MIP images showing 100× magnification of nc82 staining merged with object maps (yellow) identifying synaptic density in w[1118] and Ssdp[2082-G4]/+ brains in the SLP and SEZ regions. Scale bar = 10 μm. (F) Synaptic density is unchanged in the SLP regions ($p = 0.060$, $n = 9–10$); however, it is increased in the SEZ region ($p = 0.015$, $n = 5$) of Ssdp[2082-G4]/+ flies compared to controls. (G) MIP images showing glial staining with anti-Repo antibody in w[1118] and Ssdp[2082-G4]/+ larval CNS and adult brains. Scale bar = 100 μm. (H) Number of glial cells are reduced in both larval CNS ($p = 0.027$, $n = 9–10$) and adult brains ($1.1 \times 10^{-04}$, $n = 10–11$) of Ssdp[2082-G4]/+ flies compared to controls. Horizontal red line represents the mean value of the scatter. *P*-values are calculated using the permutation *t* test; *n* represents the number of brains analyzed per genotype. The raw data underlying panels 4B, D, F, and H can be found in S1 Data file. CNS, central nervous system; MIP, maximum intensity projection; SEZ, subesophageal zone; SLP, superior lateral protocerebrum; Ssdp, sequence-specific single-stranded DNA-binding protein.

another neuropil marker, anti-nc82 (anti-Bruchpilot), which specifically labels presynaptic active zones (Fig 4C and 4D and S5 and S6 Movies). We investigated the identity of Ssdp-positive cells in adult *Drosophila* brains and found that *Ssdp* colocalized with neuronal cell bodies (S2A Fig) but not with glial cells (S2B Fig).

Additionally, we indirectly estimated synaptic density, by using automated morphometric analysis of nc82 staining in the SLP and SEZ regions of the Ssdp[2082-G4] flies and w[1118] controls, as described earlier by Rai and colleagues [48], since SLP and SEZ are the brain regions where Ssdp is primarily expressed. The synaptic density was significantly increased in the SEZ region of Ssdp[2082-G4] flies compared to controls, and although it was decreased in the SLP region, it was not statistically significant (Fig 4E and 4F). We then analyzed the number of glia in the CNS of third instar larvae and brains of adult flies. We observed that the number of glia, labeled by anti-Repo, were significantly reduced in both Ssdp[2082-G4] larval CNS and adult brains compared to controls (Fig 4G and 4H and S7–S10 Movies).

Similarly, we then investigated the consequences of pan-neuronal *Ssdp* knockdown on brain development and glial numbers. We first analyzed the mRNA expression levels of *Ssdp* in 3- to 4-day-old pan-neuronal *Ssdp* knockdown flies and observed a significant reduction in comparison to the genotypic controls (Fig 5A). The larval CNS volume labeled by anti-DLG antibody was not different upon pan-neuronal *Ssdp* knockdown compared to controls (Fig 5B and S11 and S12 Movies). Interestingly, in contrast to *Ssdp* overexpression, pan-neuronal *Ssdp* down-regulation using Ssdp-RNAi in 3- to 4-day-old adult flies, led to smaller neuropil brain volume (Fig 5C and S13 and S14 Movies). No change was observed in the glial cell count of *Ssdp* knockdown larval CNS compared to controls (Fig 5D and S15 and S16 Movies); however, in the adult brains, the glial cell count was significantly reduced (Fig 5E and S17 and S18 Movies). Together, these results suggest that *Ssdp* plays a crucial role during neurodevelopment, and its overexpression and knockdown affects neuropil volume and glial number in *Drosophila*.

Given the neuro-anatomical defects observed in *Ssdp* overexpressing and knockdown flies, we sought to determine whether these defects are associated with defective canonical Wnt signaling known to regulate normal brain development [49–51]. Western blot analysis of 3- to 4-day-old flies revealed that armadillo expression (ortholog of human β-catenin) was significantly reduced in Ssdp[2082-G4] heads in comparison to w[1118] controls, but wingless (ortholog of human Wnt-1) expression was unaltered (Fig 6A). The *Drosophila* wing is an excellent model system for investigation of alterations in signaling pathways during development [21,52]. In contrast to a previous study [13], we observed no phenotypic defects in wing patterning but a significant decrease in wing width in Ssdp[2082-G4] compared to controls (Fig 6B). We then assessed the intensities of armadillo and wingless fluorescence in the wing discs of third instar Ssdp[2082-G4] and w[1118] larvae. We observed a significant decrease in armadillo expression in the midline area of Ssdp[2082-G4] larvae compared to w[1118] larvae, but not in the hinge line and dorsal pouch line (Fig 6C). We also observed a significant decrease in wingless expression in both dorsal pouch line and midline of in Ssdp[2082-G4] larvae compared to w[1118] larvae (Fig 6D). Interestingly, we observed a significant increase in the number of wingless-positive stripes (Fig 6D), but not in armadillo-positive stripes (Fig 6C), in the wing discs of Ssdp[2082-G4] compared to controls, which may be a compensatory mechanism for the decrease in overall wingless intensity. Similar to the adult wings, we observed a significant reduction in the width of the wing pouch in wing discs from third instar Ssdp[2082-G4] larvae compared to controls (Fig 6E). Our data hints towards the regulation of head and wing development by Ssdp through its role in the Wnt/β-catenin signaling.

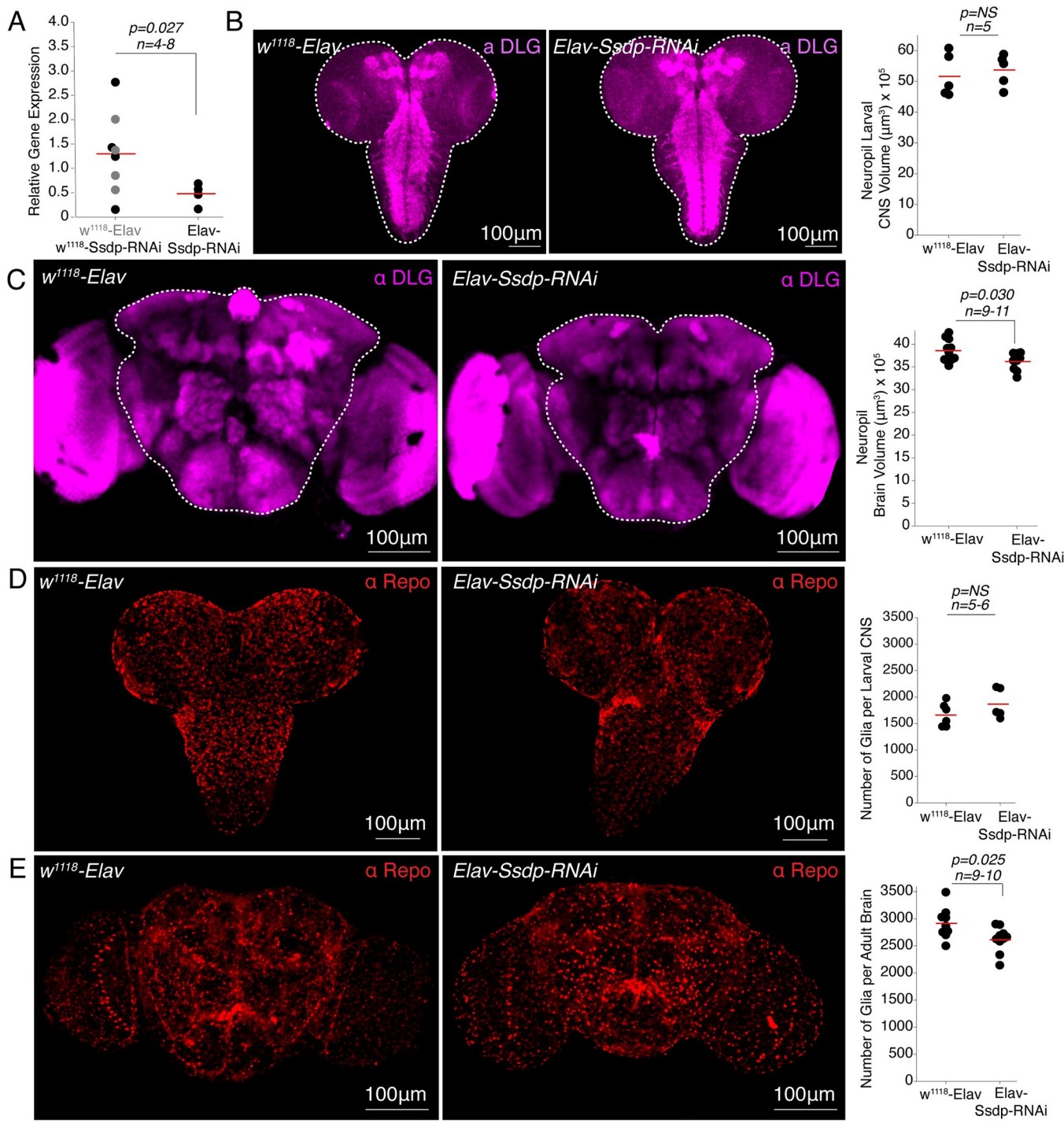

**Fig 5. *Ssdp* pan-neuronal knockdown affects adult brain development.** (A) Relative *Ssdp* mRNA expression is reduced in Elav-Gal4>UAS-Ssdp-RNAi flies compared to w[1118]>Elav-Gal4 and w[1118]>UAS-Ssdp-RNAi genotypic controls ($p = 0.027$, $n = 4$–8). Each black dot is a data point representing an independent biological replicate with 20 individual fly heads per biological replicate. Total cDNA concentration was normalized to endogenous *Rpl32* expression. *P*-values are calculated using the Student's *t* test. (B) MIP images showing larval CNS neuropils labeled by anti-DLG antibody in Elav-Gal4>UAS-Ssdp-RNAi and w[1118]>Elav-Gal4 controls. White dashed outline depicts region used to analyze CNS volume. Larval neuropil CNS volume is unchanged in *Ssdp* knockdown larvae compared to controls ($p = 0.69$, $n = 5$). (C) MIP images showing brain neuropils labeled by anti-DLG antibody in Elav-Gal4>UAS-Ssdp-RNAi adult flies and w[1118]>Elav-Gal4 controls. White dashed outline depicts central brain region used to analyze brain volume. Neuropil brain volume is significantly decreased ($p = 0.030$, $n = 9$–11) in *Ssdp* knockdown flies compared to controls. (D) MIP images showing glial staining with anti-Repo antibody in Elav-Gal4>UAS-Ssdp-RNAi and w[1118]>Elav-Gal4 larval CNS. Number of glial cells are unchanged between Elav-Gal4>UAS-Ssdp-RNAi and w[1118]>-Elav-Gal4 larval CNS ($p = 0.21$, $n = 5$–6). (E) MIP images showing glial staining with anti-Repo antibody in Elav-Gal4>UAS-Ssdp-RNAi adult flies and w[1118]>Elav-Gal4

controls. Number of glial cells are reduced in *Ssdp* knockdown flies compared to controls ($p$ = 0.025, $n$ = 9–10). Scale bar = 100 μm. Horizontal red line represents the mean value of the scatter. *P*-values are calculated using the permutation *t* test; *n* represents the number of brains analyzed per genotype. The raw data underlying this figure can be found in S1 Data file. CNS, central nervous system; MIP, maximum intensity projection; Ssdp, sequence-specific single-stranded DNA-binding protein.

## *Ssdp* overexpression causes widespread gene expression changes in *Drosophila* head

Ssdp is a DNA-binding protein and regulates gene transcription by binding to pyrimidine-rich single-stranded DNA [53]. Because of its dosage-sensitive nature, its up-regulation may interfere with its function by either dysregulating downstream gene expression or altering recruitment of proteins after binding to the DNA. Thus, we analyzed global gene expression in the heads of 3- to 4-day-old Ssdp[2082-G4] flies, by RNA sequencing (RNA-seq). Correlation matrix analysis revealed segregation of the different genotypes and high correlation between biological replicates. We first checked the expression levels of *Ssdp* mRNA and found 1.5-fold up-regulation (*p*-value <0.0001) when 3 biological replicates of Ssdp[2082-G4] were compared with 3 biological replicates of w[1118] (S2 Table). Using *Ssdp* fold change of 1.5 as the criterion (Log2 fold change ≥0.58 and p.adj ≤0.05), we found 256 down-regulated genes and 160 up-regulated genes (Fig 7A and 7B and S2 Table), among which 233 down-regulated genes and 137 up-regulated genes were protein-coding (Fig 7C).

To understand the functional implication of transcriptional dysregulation in the heads of Ssdp[2082-G4] flies, we performed gene category enrichment analysis using g:Profiler (https://biit.cs.ut.ee/gprofiler/gost). Genes that were down-regulated were highly enriched for the molecular function GO terms oxidoreductase activity and metal ion binding, which may be crucial for maintaining DNA stability and structure [54] (Fig 7D). In the up-regulated genes, serine-type endopeptidase activity (and related pathways), endopeptidase activity, monooxygenase activity, and oxidoreductase activity scored high among the molecular function GO terms (Fig 7E). Proteolysis, lactone biosynthesis processes, and immune response-related biological process GO terms were enriched in up-regulated genes (Fig 7F); however, down-regulated genes were not enriched for any brain function-related biological process GO terms. We also analyzed KEGG pathways and found glutathione metabolism (KEGG:00480, *P*-value-adj = $4.2 \times 10^{-06}$) (S3 Fig) as well as Toll and Imd signaling pathway (KEGG:04624, *P*-value-adj = $4.3 \times 10^{-02}$) (S4 Fig) in down-regulated genes, and FoxO signaling pathway (KEGG:04068, *P*-value-adj = $6.3 \times 10^{-02}$) (S5 Fig) in up-regulated genes, to be highest in significance. We also found 2 down-regulated genes (*gskt* and *sinah*) that were related to canonical Wnt signaling (S6 Fig). Interestingly many fly immunity-related genes were up-regulated (*PGRP-LD*, *Def*, *BomS5*, *BomS6*, *BomT1*, *BomT2*, *BomBc3*, *Vajk-1*, *CG18557*, *CG11842*, *IM14*), and a few immunity-related genes like *AttacinA*, *LysozymeX*, *Nazo*, *IM18*, and *PGRP-SD* were down-regulated, suggesting differential recruitment of Toll receptor-mediated signaling in *Ssdp* overexpressing flies.

We next analyzed human orthologs of dysregulated genes and found human orthologs in 65.4% (89) of up-regulated genes and 56% (143) of down-regulated genes in the heads of Ssdp[2082-G4] flies (S2 Table). Among these orthologs 36.8% of up-regulated (50) and 61% (87) of down-regulated genes had orthology scores ≥0.33 (S2 Table). We then performed a closer inspection of *Ssdp* overexpression-mediated differentially expressed genes (DEGs), as cataloged in the SFARI database. When we compared human orthologs of DEGs obtained from the heads of Ssdp[2082-G4] flies, with high-confidence ASD risk genes (Category >2) listed at SFARI database (https://gene.sfari.org/database/human-gene/, 786 genes), we found overlap with 9 (out of 87) down-regulated (*CAT*, *DNAH10*, *DNAH3*, *GPD2*, *GRIA1*, *GRID1*, *ODF3L2*,

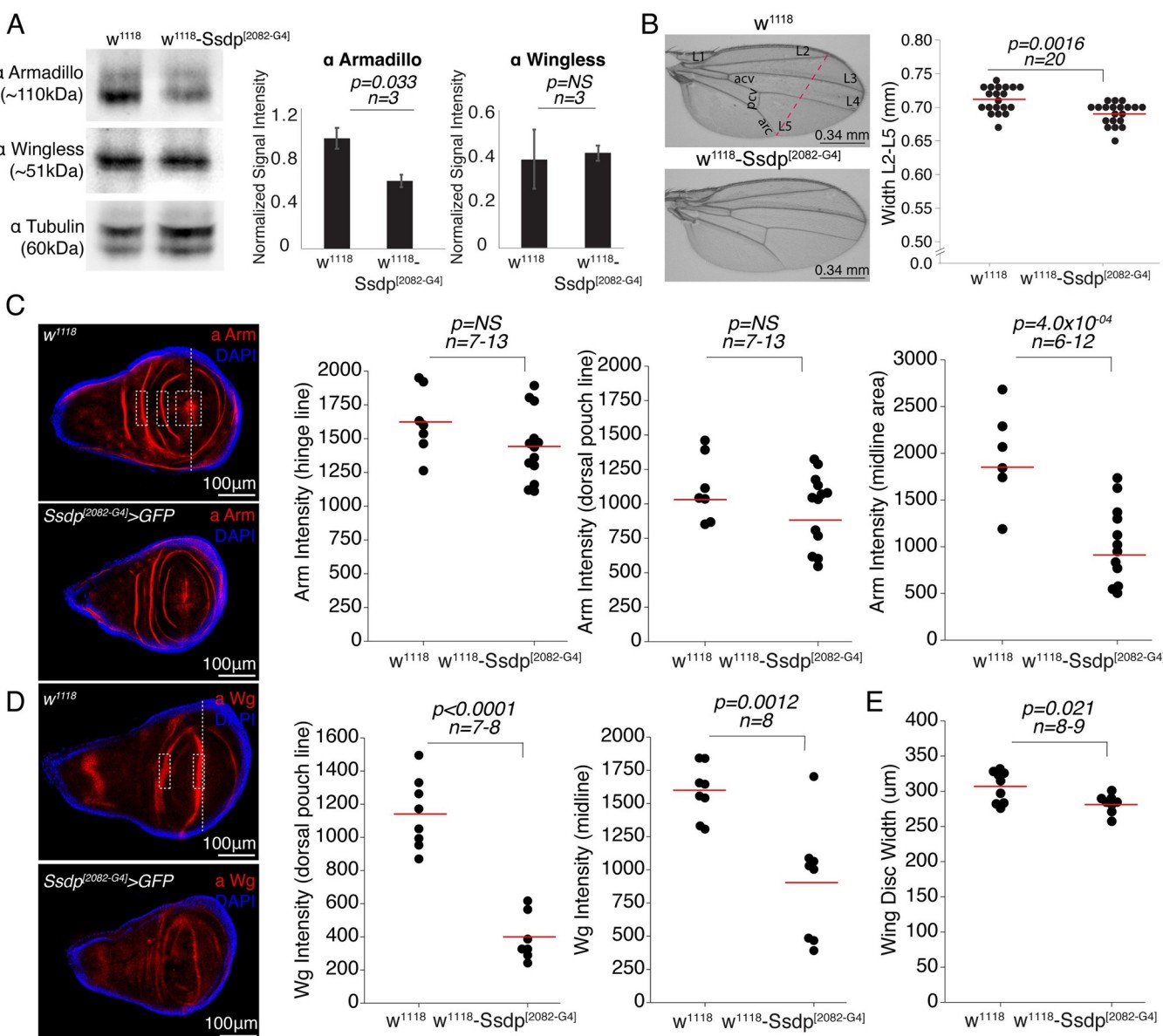

**Fig 6. *Ssdp* influences head and wing development.** (A) Armadillo is significantly reduced in the heads of heterozygous Ssdp[2082-G4]/+ compared to w[1118] controls ($p = 0.033$, $n = 3$); however, wingless expression was unaffected ($p = 0.86$, $n = 3$). (B) The wing width (L2-L5) is significantly decreased in Ssdp[2082-G4]/+ flies ($p = 0.0016$, $n = 20$) compared to controls. Pink dotted line represents the width parameter used for analysis (scale bar = 0.34 mm). (C) Wing discs showing the expression pattern of armadillo (red) and DAPI (blue) in Ssdp[2082-G4]/+ and controls. Armadillo intensity is significantly decreased in the midline area ($p = 4.0 \times 10^{-04}$, $n = 6$–12) in Ssdp[2082-G4]/+ compared to controls, but not in the hinge line ($p = 0.14$, $n = 7$–13) and the dorsal pouch line ($p = 0.22$, $n = 7$–13). White dashed rectangles show armadillo-positive hinge line, dorsal pouch line, and midline area used for intensity analysis from left to right. (D) Wing discs showing the expression pattern of wingless (red) and DAPI (blue) in Ssdp[2082-G4]/+ and controls. Wingless intensity is significantly reduced in dorsal pouch line ($p < 0.0001$, $n = 7$–8) and midline ($p = 0.0012$, $n = 8$) in Ssdp[2082-G4]/+ compared to controls. White dashed rectangles show wingless-positive dorsal pouch line and midline used for intensity analysis from left to right. (E) The width of the wing pouch is significantly decreased in Ssdp[2082-G4]/+ compared to controls ($p = 0.021$, $n = 8$–9). Vertical white dashed line represents the width measurement. Scale bar = 100 μm in the immunofluorescence images. In scatter plots, each dot represents the mean value for a single wing disc. Horizontal red line represents the mean value. *P*-values are from a two-sided permutation *t* test. In bar graphs, error bars are 95% confidence intervals. *P*-values are from Student's *t* test. The raw data underlying this figure can be found in S1 Data file. Ssdp, sequence-specific single-stranded DNA-binding protein.

*SLC6A4*, *TMLHE*) and 2 (out of 50) up-regulated genes (*GRID2* and *IGF1*). Some of these genes are highlighted in the top list of DEGs having strong orthology scores (Fig 7G). For the genes with orthology score ≥0.33, we checked the number of patients listed in DECIPHER for

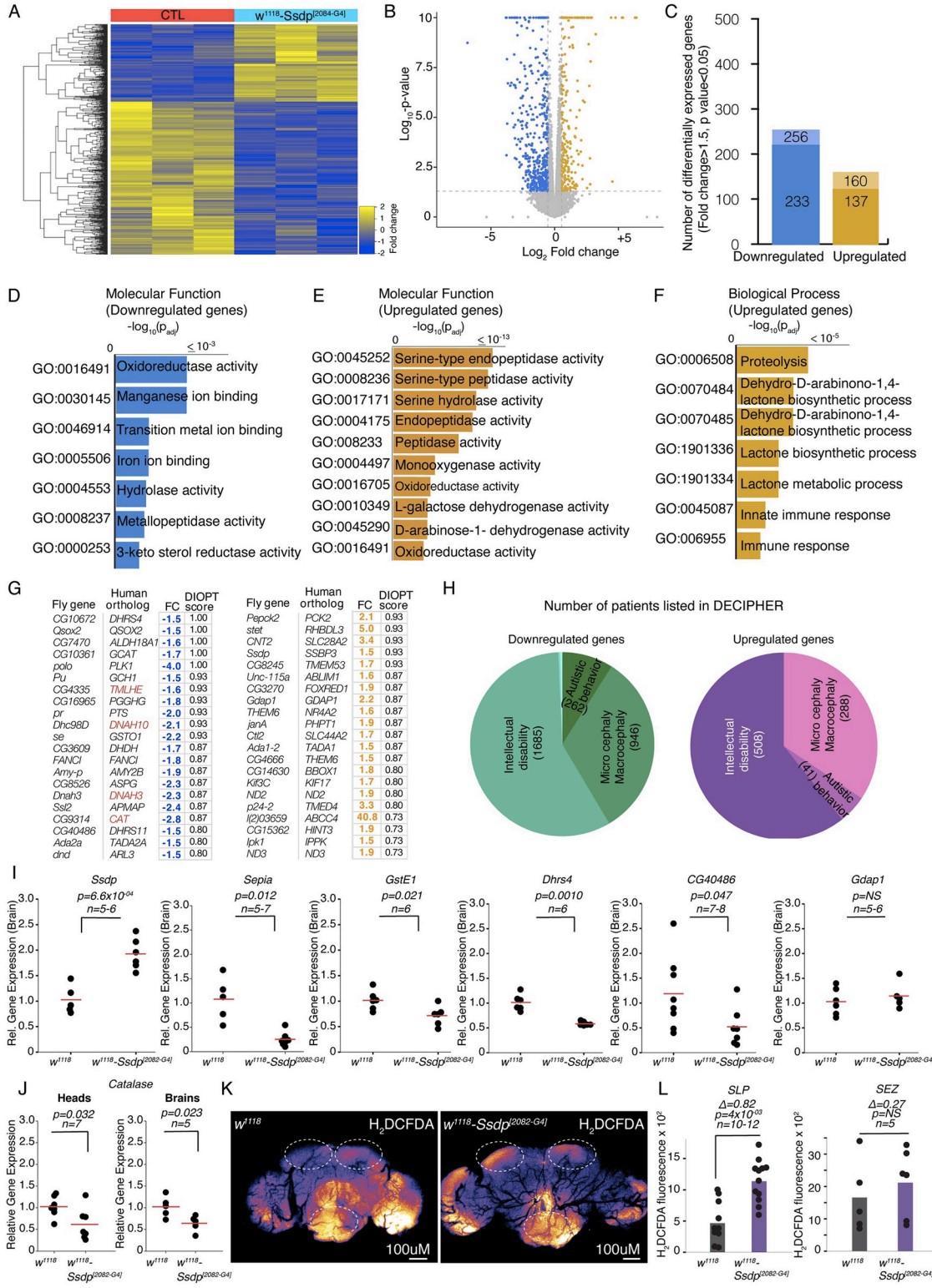

**Fig 7. Up-regulation of *Ssdp* results in alteration in transcriptomics in *Drosophila* head.** (A) Heatmap representation of 670 candidate genes identified as statistically significant (p-adj <0.05) conducted on the entire RNA-seq dataset from 3 biological replicates of heterozygous Ssdp[2082-G4]/+ heads compared to 3 biological replicates of w[1118] heads. (B) Volcano plot of differentially regulated genes in heads of Ssdp[2082-G4]/+ compared to w[1118] heads, shown as statistical significance as a function of fold change (log2 fold change >0.5 and FDR-adjusted *p*-value <0.05 are highlighted). Up-regulated genes (golden) or down-regulated genes

(blue). (C) Number of differentially regulated genes, light color indicates all gene transcripts as up- or down-regulated genes, dark colors indicate number of protein-coding genes. (D–F) GO enrichment analysis of up- and down-regulated genes. (G) List of top DEGs with strong orthology score, and highlighted in red are overlaps with high confidence ASD risk genes listed at SFARI database. (H) Overlaps with the number of patients listed at DECIPHER affected with ID, micro/macrocephaly, and autistic behaviors associated with genes differentially expressed in heads of Ssdp[2082-G4]/+ flies with an orthology score ≥0.33. (I) Validation of some DEGs using qRT-PCR from the brains of w[1118] controls and Ssdp[2082-G4]/+ experimental flies. Significant up-regulation is observed in *Ssdp* ($p = 6.6 \times 10^{-04}$, $n = 5–6$), while significant down-regulation is observed in *Sepia* ($p = 0.012$, $n = 5–7$), *GstE1* ($p = 0.021$, $n = 6$), *Dhrs4* ($p = 0.0010$, $n = 6$), and *CG40486* ($p = 0.047$, $n = 7–8$) compared to controls. While *Gdap1* ($p = 0.49$, $n = 5–6$) mRNA levels were up-regulated, they were not statistically significant. (J) Antioxidant gene *catalase* is significantly decreased in both heads ($p = 0.032$, $n = 7$) and brains ($p = 0.023$, $n = 5$) of Ssdp[2082-G4]/+ flies compared to w[1118] controls. For the qRT-PCR experiments, data are shown as mean (red horizontal line). Each black dot is a data point representing an independent biological replicate with 20 individual fly heads per biological replicate. Total cDNA concentration was normalized to endogenous *Rpl32* expression. *P*-values were obtained by Student's *t* test. (K) Pseudocolor representative images of w[1118] and Ssdp[2082-G4]/+ brains stained with H$_2$DCFDA. White dashed ovals mark regions of interest in the SLP and SEZ. (L) H$_2$DCFDA fluorescence is significantly increased in SLP per hemisphere (Cliff's Δ = 0.82 [95 CI 0.383, 0.967], $p = 4 \times 10^{-03}$, $n = 10–12$) in Ssdp[2082-G4]/+ flies in comparison to controls, with no change in the SEZ (Cliff's Δ = 0.27 [95 CI −0.6, 0.867], $p = 0.478$, $n = 5$). *P*-values are from a two-sided permutation *t* test. The raw data underlying panels 7I, J, and L can be found in S1 Data file. ASD, autism spectrum disorder; DEG, differentially expressed gene; ID, intellectual disability; SEZ, subesophageal zone; SLP, superior lateral protocerebrum; Ssdp, sequence-specific single-stranded DNA-binding protein.

micro/macrocephaly, autistic behavior, and ID and found that a large number of patients affected with these phenotypes have an association with the genes which are either down-regulated or up-regulated in the heads of Ssdp[2082-G4] flies (Fig 7H and S3 and S4 Tables).

Our RNA-seq data allowed identification of alteration in genes related to oxidoreductase, monooxygenase activity, and a decrease in glutathione metabolism that are related to oxidative stress metabolism [55–57]. Among the dysregulated genes in oxidoreductase pathway, *catalase* (an antioxidant), *sepia* (ortholog of human glutathione-s-transferase), 3 glutathione S transferases (*GstE11*, *GstE1*, and *GstD5*, all related to human *GSTT2B*), *CG9920* (ortholog of human *HSPE1*), and 2 genes of antioxidant function (*Dhrs4*, *CG40486*) were down-regulated, and *Gdap1* and *Hsp23* were up-regulated. We confirmed the mRNA levels of some of the dysregulated genes by performing qRT-PCR from the brains of 3- to 4-day-old Ssdp[2082-G4] flies and controls. We found similar trends corroborating our RNA-seq data, with statistically significant increase in *Ssdp*, and decrease in *Sepia*, *GstE1*, *Dhrs*, and *CG40486* (Fig 7I). While *Gdap1* mRNA levels were increased, they were not statistically significant (Fig 7I). qRT-PCR also confirmed a significant reduction in *catalase* in both the heads and brains of Ssdp[2082-G4] flies compared to controls (Fig 7J). To determine the levels of oxidative stress, we stained Ssdp[2082-G4] and w[1118] control brains with H$_2$DCFDA dye, an indicator of ROS. We observed a significant increase in ROS in the SLP of the Ssdp[2082-G4] brains compared to controls (Fig 7K and 7L). However, in the SEZ region, there was no significant difference (Fig 7K and 7L). Our findings provide additional evidence to support the RNA-seq data, indicating disrupted pathways related to oxidative stress as a result of *Ssdp* overexpression.

## *Ssdp* affects mitochondrial morphology and fission/fusion machinery in *Drosophila* brain

Mitochondria are dynamic and continuously undergo fission and fusion processes to maintain their shape, quality, and function. However, this machinery may be impaired in cells under stress [58]. Given the alterations in genes in RNA-seq data were associated with oxidative stress regulation (*catalase*, *sepia*, *GstE11*, *GstE1*, *GstD5*. *CG9920*), antioxidant function (*Dhrs4*, *CG40486*), and mitochondrial dynamics and function (*Gdap1* and *Hsp23*) and increase in oxidative stress in specific regions of the brain in Ssdp[2082-G4] flies, we further investigated whether these are associated with abnormalities in the morphology of mitochondria in neurons. We assessed the area, circularity, and length of mitochondria in the brains of 3- to 4-day-

old flies expressing UAS-Mito-Red in Ssdp[2082-G4]-Gal4 and Elav-Gal4 (serving as controls) (Fig 8A). Notably, we observed a difference in the pattern of Mito-Red expression using the 2 drivers, with the distribution being more spread and diffuse in the control brains. We suggest that this is an artifact of the Elav-Gal4 driver rather than a representation of mitochondrial morphology [59,60]. Hence, the SEZ region, presenting a more comparable mitochondrial morphology, was used for analysis in the controls. We observed a significant decrease in the area and circularity of mitochondria in both SLP and SEZ (Fig 8B). However, the length was significantly increased (Fig 8B). This data suggests an enhanced mitochondrial fusion or defective fission process [51] in *Ssdp* overexpressing flies. We then performed qRT-PCR to investigate genes involved in the fission/fusion process in the heads of Ssdp[2082-G4] and controls and observed the up-regulation of 1 fission (*Drp1*) and 1 fusion gene (*Marf*) (Fig 8C). The expression levels of *Fis1* and *Opa1* were not affected. We further confirmed the up-regulation of *Drp1* and *Marf* by performing qRT-PCR on the brains of 3- to 4-day-old Ssdp[2082-G4] and controls (S7 Fig). Overall, our data suggests that *Ssdp* overexpression results in disbalance of mitochondrial fission/fusion machinery, with an enhancement in fusion events.

## Alteration in *Ssdp* expression affects anxiety-like behavior and decision-making

Anxiety and difficulty in decision-making are common phenotypes observed in autistic individuals [49,50,52]. Multiple high-confidence ASD risk genes associated with anxiety were dysregulated in our RNA-seq data including *TMLHE* and *GRID2* [61], *DNAH10* [62], *GRIA1* [63], *SLC6A4* [64], and *IGF1* [65]. Further, genes associated with motor defects were also dysregulated including *GRIA1* [66], *SLC6A4* [67], and *GRID2* [68]. To investigate the effects of *Ssdp* CNVs on fly locomotor behavior, we recorded the activity of 3- to 4-day-old flies using the *Drosophila* arousal tracking (DART) system in an open field arena [69]. In this assay, apart from measuring fly locomotor output like average walking speed, we also measured wall-following, an anxiety-like behavior [19,69]. We observed that *Ssdp* manipulations did not affect average locomotor speed (Fig 9A). However, Ssdp[2082-G4] flies but not the knockdown flies, exhibited a significant increase in the percentage of time spent near the wall edge compared to the controls, suggesting higher anxiety (Figs 9B and S8A).

We next assessed wall approach-avoidance behavior in an open field [69], by analyzing the proportional distribution of the wall approach angles of 3- to 4-day-old flies. A fly approaching the wall avoids a head-on collision [69]; however, a fly defective in quick decision-making might collide with the wall more often. As reported earlier [69], control flies exhibited a bimodal distribution in wall approach angles, with the most preferred angles being around ±45˚ (Fig 9C). In contrast to the bimodal distribution of control flies, both *Ssdp* knockdown and *Ssdp* overexpressing flies showed flattened approach-angle distribution, with a peak around 0˚ (Fig 9C). These results suggest that flies with altered *Ssdp* expression experience indecisiveness when approaching the wall and walk head-on, which might cause head-butting. Overall, our data indicates that *Ssdp* regulates mental processes including anxiety-like behaviors and decision-making.

## *Ssdp* affects sensory perception and habituation

Adult *Drosophila* startle in response to odor [70] and light-on-light-off-stimuli [71]. Repeated presentation of the same stimuli leads to a decrease in behavioral response, known as habituation learning. Habituation learning represents a higher cognitive function of filtering and processing sensory information to navigate a dynamic environment [71]. Defective learning, or ID, is a core behavioral feature of individuals with NDDs like ASD [72] and has previously

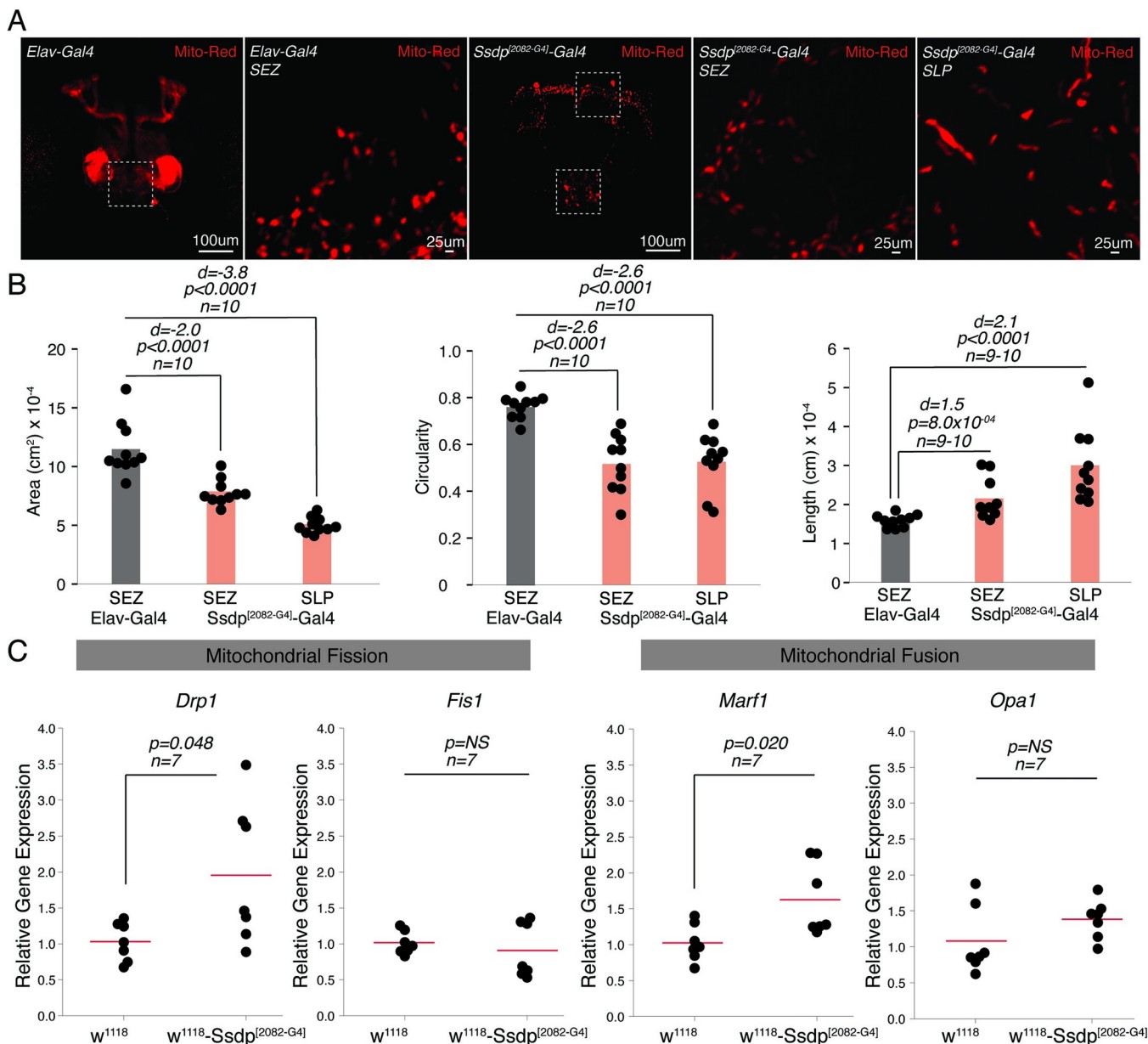

**Fig 8. *Ssdp* overexpression causes abnormalities in mitochondrial morphology and dynamics.** (A) Immunohistochemistry images of brains expressing UAS-Mito-Red in the Elav-Gal4 and Ssdp[2082-G4]-Gal4 labeled neurons. White squares highlight the SEZ and SLP in Ssdp[2082-G4]-Gal4 image. Magnified images of SEZ and SLP regions are also shown. (B) Mitochondrial area in SEZ (Cohen's d = −2.0 [95 CI −2.84, −1.13], $p < 0.0001$, $n = 10$) and SLP (Cohen's d = −3.8 [95 CI −5.39, −2.55], $p < 0.0001$, $n = 10$), and circularity in SEZ (Cohen's d = −2.6 [95 CI −3.57, −1.65], $p < 0.0001$, $n = 10$) and SLP (Cohen's d = −2.6 [95 CI −3.4, −1.6], $p < 0.0001$, $n = 10$) are significantly decreased, while length in SEZ (Cohen's d = 1.5 [95 CI 0.822, 2.19], $p = 8.0 \times 10^{-04}$, $n = 9$–10) and SLP (Cohen's d = 2.1 [95 CI 1.29, 2.86], $p < 0.0001$, $n = 10$) is significantly increased in Ssdp[2082-G4] in comparison to Elav controls. Quantification was performed per hemisphere. (C) Genes associated with mitochondrial fission (*Drp1*, $p = 0.048$, $n = 7$) and mitochondrial fusion (*Marf1*, $p = 0.020$, $n = 7$) are up-regulated in Ssdp[2082-G4]/+ in comparison to w[1118] controls. *Fis1* ($p = 0.54$, $n = 7$) and *Opa1* ($p = 0.16$, $n = 7$) mRNA levels were unaltered in Ssdp[2082-G4]/+ flies compared to controls. *P*-values were obtained by Student's *t* test. For all qRT-PCR experiments, expression was normalized to *Rpl32*. Horizontal red line represents the mean value. Each black dot is a data point representing an independent biological replicate. For B, *p*-values were obtained by permutation *t* test. *P*-values less than 0.05 were considered significant. The raw data underlying panels 8B and C can be found in S1 Data file. SEZ, subesophageal zone; SLP, superior lateral protocerebrum; Ssdp, sequence-specific single-stranded DNA-binding protein.

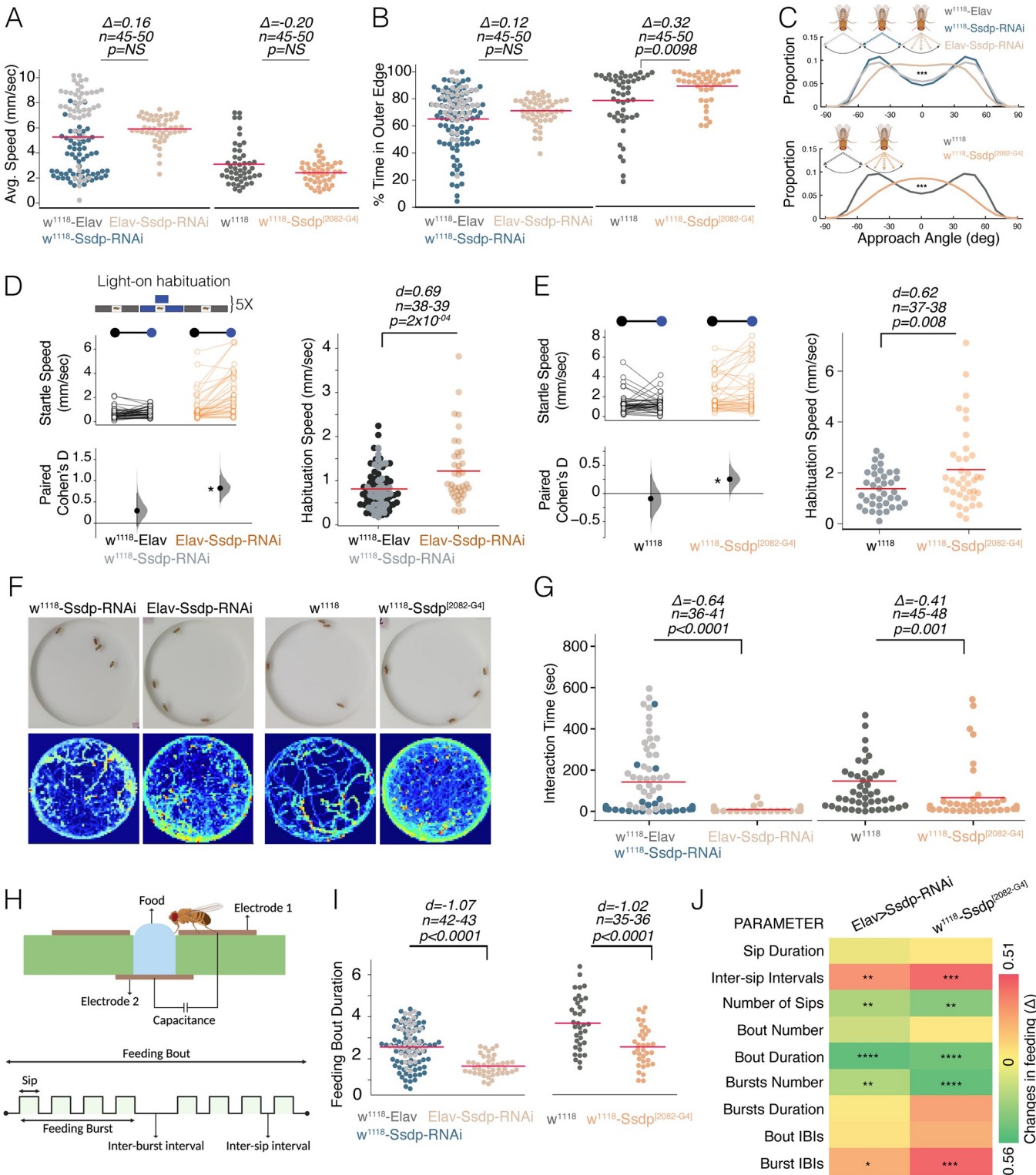

**Fig 9. *Ssdp* knockdown and overexpression cause autism-like behavioral deficits in *Drosophila*.** (A) Active average speed in an open field arena is unaffected in Elav-Gal4>UAS-Ssdp-RNAi flies compared to the genotypic controls w[1118]>Elav-Gal4 and w[1118]>Ssdp-RNAi (*n* = 45–50, Cliff's Δ = 0.16 [95 CI −0.0242, 0.335], *p* = 0.12), and in heterozygous Ssdp[2082-G4]/+ flies compared to w[1118] controls (*n* = 45–50, Cliff's Δ = −0.20 [95 CI −0.427, 0.0407], *p* = 0.0938). (B) Ssdp[2082-G4]/+ (*n* = 45–50, Cliff's Δ = 0.32 [95 CI 0.0787, 0.527], *p* = 0.0098) but not knockdown flies (*n* = 45–50, Cliff's Δ = 0.12 [95 CI −0.0702, 0.293], *p* = 0.256) show a significantly higher wall-following compared to controls. (C) *Ssdp* knockdown and Ssdp[2082-G4] flies are impaired in approach angle bimodal distribution. Fly representations created with BioRender.com. (D) *Ssdp* knockdown flies show increased startle to the first blue light pulse (genotypic controls,

$n = 38$, Cohen's d = 0.48 [95 CI 0.28, 0.69], $p < 0.0001$; Elav>Ssdp-RNAi, $n = 38$, Cohen's d = 0.8 [95 CI 0.53, 1.1], $p < 0.0001$) and defective habituation to the fifth pulse, compared to their genotypic controls ($n = 37–39$, Cohen's d = 0.69 [95 CI 0.31, 1.1], $p = 2.0 \times 10^{-04}$). (E) Ssdp[2082-G4]/+ also show increased startle to the first pulse (w[1118], $n = 39$, Cohen's d = −0.09 [95 CI −0.43, 0.37], $p = 0.7$; Ssdp[2082-G4]/+, $n = 38$, Cohen's d = 0.3 [95 CI 0.07, 0.55], $p = 0.025$) and defective habituation to the fifth pulse compared to controls ($n = 37–38$, Cohen's d = 0.62 [95 CI 0.23, 0.96], $p = 0.008$). (F) Sample frames of interactions among males of different genotypes (top) and heatmap occupancy plots for all flies in each group (bottom). (G) Interaction time was reduced in both *Ssdp* knockdown ($n = 36–41$, Cliff's Δ = −0.64 [95 CI −0.79, −0.44], $p < 0.0001$) and Ssdp[2082-G4]/+ flies ($n = 45–48$, Cliff's Δ = −0.41 [95 CI −0.62, −0.17], $p = 0.001$). (H) Schematic of FlyPad assay. Created with BioRender.com. (I) *Ssdp* knockdown ($n = 42–43$, Cohen's d = −1.07 [95 CI −1.39, −0.73], $p < 0.0001$) and Ssdp[2082-G4]/+ ($n = 35–36$, Cohen's d = −1.02 [95 CI −1.5, −0.57], $p < 0.0001$) flies exhibit reduced duration of interaction with food. (J) Heatmap depicts effect size (Cliff's Δ) for all the feeding parameters between controls and experimental groups. Asterisks indicate level of statistical significance: $^*p \leq 0.05$, $^{**}p \leq 0.01$, $^{***}p \leq 0.001$, $^{****}p \leq 0.0001$. In scatter plots, each dot represents the mean value for a single fly. Horizontal red line represents the mean value. *P*-values are from a two-sided permutation *t* test. The raw data underlying panels 9A–E, G, and I can be found in S1 Data file. Ssdp, sequence-specific single-stranded DNA-binding protein.

been used to characterize autism genes in *Drosophila* [71]. To investigate the functional role of *Ssdp* in startle and habituation behavior, we subjected 3- to 4-day-old flies to a blue light-on after-dark period paradigm, measuring startle response to a visual stimulus and habituation to repeated stimuli. *Ssdp* knockdown and Ssdp[2082-G4] flies displayed enhanced baseline speed compared to controls (S8B and S8C Fig, respectively) and heightened startle response to the first blue light pulse (Fig 9D and 9E, respectively). *Ssdp* knockdown and Ssdp[2082-G4] flies also exhibited defective habituation to the fifth light pulse compared to the controls (Fig 9D and 9E, respectively). This data provides the evidence that *Ssdp* is required by *Drosophila* for visual filtering mechanisms and processing sensory information. From our RNA-Seq data, among the 11 differentially regulated high-confidence ASD risk genes, there are few, which display learning disability phenotypes, which include *GRIA1* [73], *GRID2* [74,75], and *IGF1* [76].

## Changes in *Ssdp* expression impairs social interactions

*Drosophila* provides a range of behavioral repertoires, including social interactions, which may be reminiscent of human social interaction behavior [77,78]. Impaired social interactions have previously been shown in multiple *Drosophila* ASD models [17,20,79,80]. Here, we studied social behaviors in a group of four 3- to 4-day-old male flies having reduced or elevated *Ssdp* expression. We analyzed parameters such as the sitting interaction time of each fly with another fly and the number of sitting contacts in 20 min. We observed that *Ssdp* knockdown and Ssdp[2082-G4] flies have significantly decreased interaction time (Fig 9F and 9G) and decreased the number of contacts (S8D Fig) with each other compared to their respective controls. Our data strongly suggests that both decreased and elevated levels of *Ssdp* cause impairment in social interactions. We inspected our RNA-seq data and found that among the 11 differentially regulated high-confidence ASD risk genes, *GRIA1* [81], *GRID1* [82], and *IGF1* [83] have been shown to modulate social behavior.

## *Ssdp* regulates feeding behavior

Feeding-related problems are very common among individuals affected with ASD [84,85]. In the *Drosophila* brain, Ssdp expresses in many SEZ neurons, a region known to affect feeding behavior [86]. We determined if the altered *Ssdp* expression might influence feeding behavior in 3- to 4-day-old flies. We utilized the Fly Proboscis and Activity detector (FlyPAD) [87] (Fig 9H) and observed that *Ssdp* knockdown and Ssdp[2082-G4] flies have significantly fewer sips and meals, with decreased duration of interaction with food and increased intervals between sips and meals (Figs 9I and 9J and S8E). Our data suggests that *Ssdp* regulates feeding behavior in a dose-dependent manner. Closer inspection of RNA-seq data revealed that hunger and satiety controlling factors were among the dysregulated genes in heads of Ssdp[2082-G4] flies. Neuropeptides such as insulin-like peptides (*ILPs*) 3 and 5 were up-regulated, and the satiety controlling neurohormone, female-specific independent of transformer (*fit*) was down-regulated.

These *ILP*s and the neurohormone, *fit*, are known to be secreted from a specific set of neurons in the fly brain and control satiety signals that negatively affect motivation for feeding [88]. It is worth noting here that in the *Drosophila* brain, Insulin Receptor (*InR*) is known to suppress the neurohormone *fit* [88].

## Partial rescue of behavioral defects upon knocking down *Ssdp* expression in *Ssdp*-overexpressing cells

Given that overexpression of *Ssdp* in Ssdp-positive cells produced multiple behavioral defects, we next asked whether knockdown of *Ssdp* in these cells would rescue the behavioral defects observed. We first performed qRT-PCR to analyze *Ssdp* mRNA expression in the heads of 3- to 4-day-old Ssdp[2082-G4] flies and Ssdp-RNAi-expressing Ssdp[2082-G4] flies and compared it to the relative mRNA expression in heads of w[1118] controls. Consistent with our previous finding, *Ssdp* mRNA levels were increased in the heads of Ssdp[2082-G4] flies; however, the *Ssdp* mRNA level in the heads of Ssdp-RNAi-expressing Ssdp[2082-G4] flies were reduced by 12.5% compared to Ssdp[2082-G4] flies but were not statistically significant (S9A Fig). Furthermore, the mRNA levels in Ssdp-RNAi-expressing Ssdp[2082-G4] flies were still higher compared to w[1118] controls but they were not statistically different (S9A Fig). We then performed multiple behavioral analyses on 3- to 4-day-old Ssdp[2082-G4] and Ssdp-RNAi-expressing Ssdp[2082-G4] flies to determine the implications of this reduction in *Ssdp*. As shown before (Figs 9 and S8), in Ssdp[2082-G4] flies, locomotor activity was unchanged (S9B Fig), wall-following was enhanced (S9C Fig), habituation speed was decreased (S9D Fig), and interaction time and number of contacts were decreased (S9E Fig) compared to controls. However, in Ssdp-RNAi-expressing Ssdp[2082-G4] flies, locomotor activity was reduced (S9B Fig), wall-following behavior was unchanged (S9C Fig), habituation speed was unchanged (S9D Fig), and sociation interaction defects were still observed (S9E Fig), in comparison to w[1118] controls. Compared to Ssdp[2082-G4] flies, Ssdp-RNAi-expressing Ssdp[2082-G4] flies showed decreased average speed (S9B Fig), anxiety (S9C Fig), and habituation speed (S9D Fig), but showed no change in social interaction (S9E Fig). Our data suggests that knocking down *Ssdp* in Ssdp-positive cells that overexpress *Ssdp*, normalizes *Ssdp* mRNA levels and rescues anxiety and habituation learning behavioral deficits. We rationalize that even miniscule alterations in Ssdp levels manifest as alterations in the social interaction of the flies.

## *Ssdp* knockdown exclusively in adult brains does not produce changes in autism-associated phenotypes

*Drosophila* genetics allow temporal regulation of gene expression with the help of the temperature sensitive Gal80 allele (Gal80[ts]). At lower temperatures, the Gal80[ts] protein binds to Gal4 and prevents gene expression, while at higher temperatures, the Gal80[ts] protein becomes non-functional, allowing normal Gal4-dependent gene expression [89]. We explored the use of Gal80[ts] to perform pan-neuronal *Ssdp* knockdown exclusively in the adult progeny, to determine whether it would be sufficient to produce behavioral and functional defects as observed before (S10A Fig). In the open field assay, overall we observed no change in average speed and wall-following; however, decisiveness was altered in Elav-Gal80[ts]-Ssdp-RNAi flies compared to w[1118]-Elav-Gal80[ts] controls (S10B Fig). In the blue light-on after-dark period paradigm, habituation speed to the fifth blue light pulse was unchanged between Elav-Gal80[ts]-Ssdp-RNAi flies and w[1118]-Elav-Gal80[ts] controls (S10C Fig). In the social interaction assay, the interaction time and number of contacts were similar between Elav-Gal80[ts]-Ssdp-RNAi flies and w[1118]-Elav-Gal80[ts] controls (S10D Fig). We further explored whether oxidative stress in the brain was altered upon *Ssdp* knockdown exclusively in adult flies. We observed that there was no change in the H$_2$DCFDA fluorescence between Elav-

Gal80[ts]-Ssdp-RNAi flies and w[1118]-Elav-Gal80[ts] controls in both SLP and SEZ regions (S10E Fig). Our data shows that pan-neuronal *Ssdp* knockdown in adult and developed brains are not sufficient to produce behavioral and functional defects.

## Optogenetic manipulations of Ssdp-expressing neurons alter autism-associated behaviors

As the characteristics of *Drosophila* Ssdp- and human SSBP3-expressing neurons are very similar, we next asked whether manipulating neuronal activity using optogenetic actuators of Ssdp-labeled neurons would also alter autism-associated behaviors. Optogenetics is a powerful tool that allows regulation of neuronal activity in a non-invasive manner using light. We used the optogenetic activator, CsChrimson [90], and the optogenetic inhibitor, GtACR1 [91], to activate or inhibit Ssdp-expressing neurons respectively, using the GAL4/UAS system. These tools have been extensively used to investigate various kinds of behaviors in *Drosophila* such as locomotion [92], sleep [93], courtship [94], learning and memory [95], and stress-related behaviors [96]. We assessed locomotion, wall-following, habituation learning, and social interaction in 3- to 4-day-old Chrimson-Ssdp[2082-G4] flies upon red illumination and in GtACR1-Ssdp[2082-G4] flies upon green illumination and compared the behaviors to their respective genotypic controls. In the open field assay, the flies were exposed to red (635 nm, 0.3 mW/mm$^2$) or green (532 nm, 0.2 mW/mm$^2$) light for the entirety of the assay time (10 min) (Fig 10A). In the habituation assay, for activation of Ssdp-expressing neurons, the flies were exposed to red light (635 nm, 0.3 mW/mm$^2$) continuously throughout the assay time (6 min), with 1-s blue light pulses shone at 1-min intervals (Fig 10D). However, due to the blue-green sensitivity of the *Drosophila* photoreceptors [97], for inhibition of Ssdp-expressing neurons, the flies were simultaneously exposed to blue and green (532 nm, 0.2 mW/mm$^2$) light pulses for 1-s at 1-min intervals (Fig 10D). In the social interaction assay, the flies were exposed to red (635 nm, 0.08 mW/mm$^2$) or green (532 nm, 0.09 mW/mm$^2$) light for the entirety of the assay time (20 min) (Fig 10F).

As we showed above (Figs 9 and S8), Ssdp[2082-G4] flies, under normal white light illumination, had no change in average speed (Fig 10B), enhanced wall-following behavior (Fig 10C), increased habituation (Fig 10E) and startle speed (S11A Fig), and decreased social interaction time (Fig 10G) and number of contacts (S11B Fig). Upon red light illumination, Chrimson-Ssdp[2082-G4] flies showed increased average speed (Fig 10B), decreased wall-following behavior (Fig 10C), decreased habituation speed (Fig 10E) and startle speed (S11A Fig), and increased interaction time (Fig 10G) and number of contacts (S10B Fig). Upon green illumination, GtACR1-Ssdp[2082-G4] flies showed decreased average speed (Fig 10B), decreased wall-following behavior (Fig 10C), decreased habituation speed (Fig 10E) and startle speed (S11A Fig), and decreased interaction time (Fig 10G) and number of contacts (S11B Fig). Our data suggests that Ssdp-expressing neurons are both sufficient and necessary for modulating locomotion and social interaction. However, for regulation of anxiety and habituation learning, an interplay of other neuronal circuits with the Ssdp-expressing neurons may be required, since these behaviors were altered even when Ssdp-expressing neurons were optogenetically inhibited. Interestingly, optogenetic manipulation of Ssdp-expressing neurons had a positive effect over anxiety as the flies displayed decreased wall-following behavior. However, habituation learning was still impaired as the flies displayed a "freeze response" to the blue light pulse, interpreted from the decreased startle and habituation speed.

## Discussion

In this study, we investigated the role of *Ssdp* in neurodevelopment and autism-associated phenotypes. Our findings reveal that SSBP3 and its *Drosophila* ortholog Ssdp are primarily

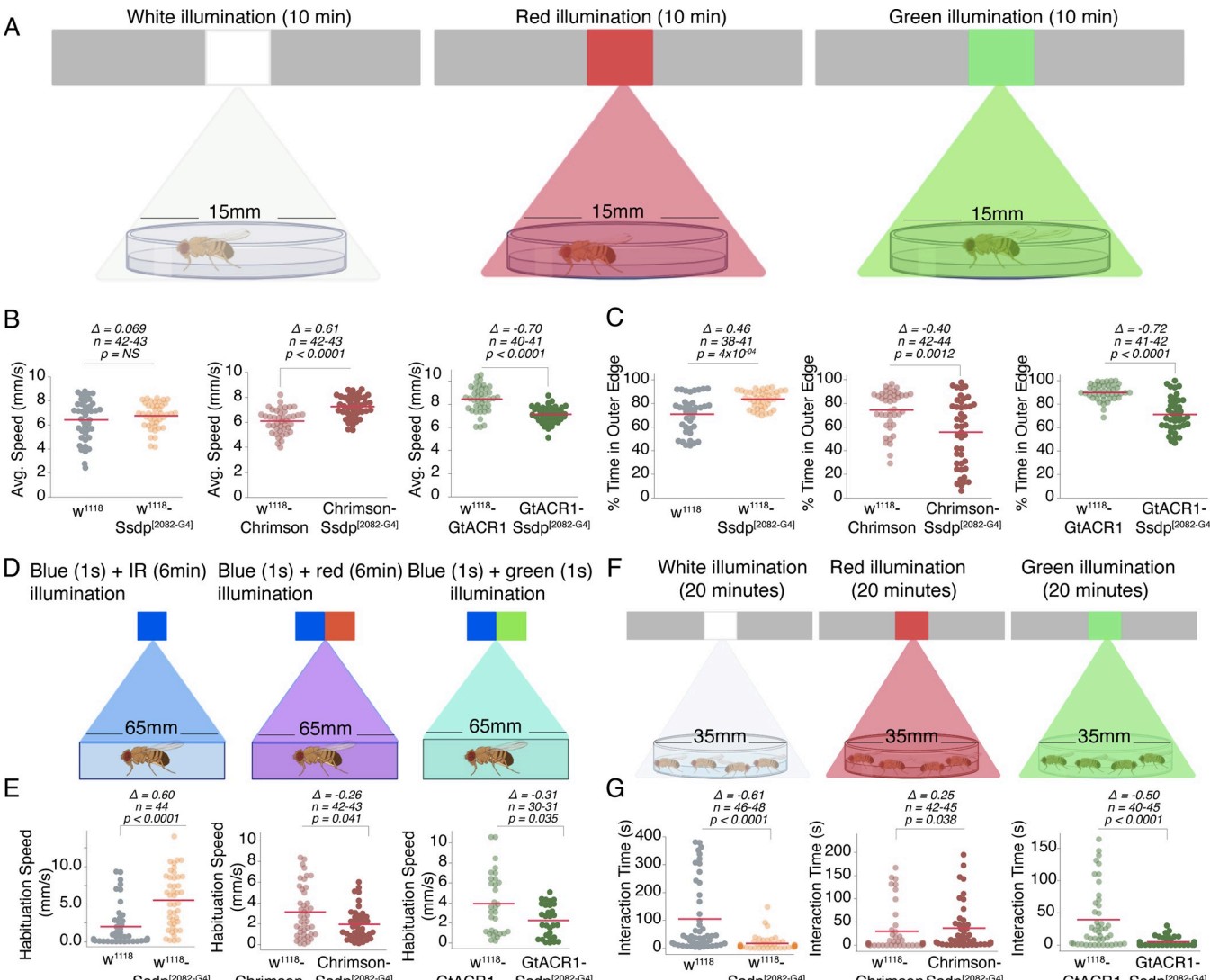

**Fig 10. Actuating neuronal activity of Ssdp-expressing neurons alters autism-associated behaviors.** (A) Schematic depicting assessment of behavior in an open field arena in the presence of white, red, or green illumination. Created with BioRender.com. (B) Active average speed is unaffected in heterozygous Ssdp[2082-G4] flies compared to w[1118] controls ($n$ = 42–43, Cliff's Δ = 0.069 [95 CI −0.185, 0.328], $p$ = 0.59); however, it is significantly increased in Ssdp[2082-G4]>UAS-Chrimson flies ($n$ = 42–43, Cliff's Δ = 0.61 [95 CI 0.382, 0.766], $p$ < 0.0001) and significantly decreased in Ssdp[2082-G4]>UAS-GtACR1 flies ($n$ = 40–41, Cliff's Δ = −0.70 [95 CI −0.835, −0.462], $p$ < 0.0001) compared to w[1118]>UAS-Chrimson and w[1118]>UAS-GtACR1 controls, respectively. (C) Wall-following behavior is significantly increased in Ssdp[2082-G4] flies compared to w[1118] controls ($n$ = 38–41, Cliff's Δ = 0.46 [95 CI 0.189, 0.673], $p$ = 4 × 10^{−04}); however, it is significantly decreased in Ssdp[2082-G4]>UAS-Chrimson ($n$ = 42–44, Cliff's Δ = −0.40 [95 CI −0.603, −0.157], $p$ = 0.0012) and Ssdp[2082-G4]>UAS-GtACR1 flies ($n$ = 41–42, Cliff's Δ = −0.72 [95 CI −0.858, −0.499], $p$<0.0001) compared to w[1118]>UAS-Chrimson and w[1118]>UAS-GtACR1 controls respectively. (D) Schematic depicting modified blue light-on after-dark period paradigm for optogenetic manipulation. Created with BioRender.com. (E) Habituation speed is significantly increased in Ssdp[2082-G4] flies ($n$ = 44, Cliff's Δ = 0.60 [95 CI 0.376, 0.756], $p$ < 0.0001); however, it is significantly decreased in Ssdp[2082-G4]>UAS-Chrimson ($n$ = 42–43, Cliff's Δ = −0.26 [95 CI −0.487, −0.00886], $p$ = 0.041) and Ssdp[2082-G4]>UAS-GtACR1 flies ($n$ = 30–31, Cliff's Δ = −0.31 [95 CI −0.563, −0.0043], $p$ = 0.035) compared to their controls. (F) Schematic depicting assessment of social interaction in the presence of white, red, or green illumination. Created with BioRender.com. (G) Interaction time is significantly decreased in Ssdp[2082-G4] flies ($n$ = 46–48, Cliff's Δ = −0.61 [95 CI −0.761, −0.414], $p$ < 0.0001) and Ssdp[2082-G4]>UAS-GtACR1 flies ($n$ = 40–45, Cliff's Δ = −0.50 [95 CI −0.679, −0.261], $p$ < 0.0001); however, it is significantly increased in Ssdp[2082-G4]>UAS-Chrimson flies ($n$ = 42–45, Cliff's Δ = 0.25 [95 CI 0.00529, 0.478], $p$ = 0.038), compared to their controls. In scatter plots, each dot represents the mean value for a single fly. Horizontal red line represents the mean value. *P*-values are from a two-sided permutation *t* test. The raw data underlying panels 10B, C, E, and G can be found in S1 Data file. Ssdp, sequence-specific single-stranded DNA-binding protein.

expressed in excitatory neurons. Interestingly, SSBP3 expression in human brain, colocalizes with *SLC17A7*, a gene from the solute carrier family that has been associated with autism [98,99], suggesting a potential link between *Ssdp* and autism-related pathways. Excitatory glutamatergic neurons in the cortex and the striatum have previously been implicated in ASD [100,101]. We also observed anomalies in bristle number and structure of the Ssdp[2082-G4] heads, which may suggest a role for Ssdp in bristle morphogenesis during development. Bristles in *Drosophila* serve as mechanosensory and chemoreceptor devices, allowing flies to detect environmental cues [102]. Therefore, the observed anomalies in bristles may have functional and behavioral consequences in the Ssdp[2082-G4] flies potentially impacting their sensory perception and behavior. Subsequent studies may investigate whether changes in *Ssdp* levels in sensory neurons or antenna are adequate to impact behaviors like social interaction, feeding, and habituation.

Furthermore, we also observed variable synaptic densities in the brains of Ssdp[2082-G4] flies resembling CNS pathologies observed in ASD patients. Synaptic dysfunction has been consistently implicated in ASD neuropathology, with several ASD-associated genes playing roles in synaptic plasticity and synaptic homeostasis [103,104].

In comparison to larger neuropil brain volumes observed in Ssdp[2082-G4] larvae and adult flies, we observed smaller neuropil brain volumes in *Ssdp* knockdown adult flies, emphasizing its role in cephalization. Such contrasting brain sizes have even been observed in ASD patients, where duplications and deletions in 16p11.2 have opposing head sizes [105,106], due to which ASD is also classified as a synaptopathy [107].

In contrast to larger brains, Ssdp[2082-G4] have significantly smaller wing discs, indicating that *Ssdp* may play an inhibitory or activatory role in a tissue context-dependent manner, similar to the mini-brain (Mnb) in brain and wing development [108]. This suggests that *Ssdp* may have complex and context-dependent functions in different tissues during development.

Given that SSBP3/Ssdp is also expressed in oligodendrocytes and glial cells in humans and *Drosophila*, respectively, the observed decrease in numbers of glial cells in *Ssdp* overexpressing larvae and adult flies and *Ssdp* knockdown flies may indicate an essential role of *Ssdp* in glial cell survival. Glial cells are known to play crucial roles in neurodevelopment, maintenance of the nervous system, pruning, and glial dysfunction has been implicated in various NDDs [109,110]. Therefore, our findings suggest that *Ssdp* may regulate gliogenesis and may in turn affect glial function. The lack of neuropil volume and glial number changes in pan-neuronal *Ssdp* knockdown larvae could be due to the difference in susceptibility of the CNS to changes in *Ssdp* levels using Elav-Gal4 driver during development, an Ssdp-Gal4 driver inserted outside the Ssdp locus maybe used for further investigation. Moreover, it further indicates the coordinated regulation of glial number and brain volume.

Upon close inspection of our RNA-seq data, we identified 5 down-regulated genes that are directly involved in different aspects of neurodevelopment. For instance, genetic variants of *SLC30A2* have been associated with zinc deficiency, which affects brain development and produces neurobehavioral impairments [111]. *TUBB2B*'s has been implicated in polymicrogyria, a condition, characterized by increased folding of cerebral cortex and cortical layer malformations [112]. Knockdown of *NAP1L1* in mice has been shown to decrease proliferation of neural progenitor cells and affect cortical neurogenesis during embryonic brain development [113]. Biallelic variants in *SORD* gene have been identified as a causal factor for axonal hereditary neuropathy [114]. *SIAH1* has been shown to be dysregulated in ASD individuals in association with axonogenesis and neurite development [115]. Therefore, our data indicates a connection between *Ssdp* and other genes that mediate brain development, neuronal networks, and processes that ultimately affect various behaviors.

We also observed reduced expression of armadillo in Ssdp[2082-G4] heads and wing discs compared to controls. Wnt/β-catenin signaling is known to play diverse roles during various

processes such as neurogenesis [116], gliogenesis [117], pruning [118], and neuroprotection [74]. Reduction in the Wnt/β-catenin signaling pathway in Ssdp[2082-G4] brains might represent the molecular mechanism underlying larger brain volume and reduced glial number [119]. Intriguingly, β-catenin signaling and truncating mutations have been reported in ASD individuals and have been shown to produce microcephaly and ASD-associated behavioral deficits such as impaired social interactions and repetitive behaviors in mice [119–122]. Analysis of KEGG pathways for differentially regulated genes upon *Ssdp* overexpression revealed down-regulation of *sinah* (ortholog of human *SIAH1)* and *gskt* (ortholog of human *GSK3β*), which are both associated with the canonical Wnt signaling pathway. Normally, SIAH1 promotes β-catenin accumulation, by ubiquitinating and degrading Axin, which along with GSK3 degrades β-catenin [123,124]. Thus, down-regulation of *sinah* may explain reduced armadillo levels in *Ssdp* overexpressing flies. However, further investigation is required to determine precise mechanism and neurodevelopment time points during which Ssdp regulates β-catenin and leads to reduced gliogenesis and larger brain phenotype.

RNA-seq also enabled us to identify differential expression of multiple genes, many of which were associated with oxidative stress-related pathways. Endopeptidase activity was enriched among the up-regulated genes, which is interesting, as it has been associated with autism-related behavioral deficits, possibly due to peptidases cleaving important behavioral neuropeptides [125,126]. Our DEGs were specifically enriched with genes that are shown to be associated with ID, micro/macrocephaly, and autistic behaviors. Many known high confidence autism-risk genes were among the DEGs, and many of the dysregulated genes were associated with impairment in anxiety, learning, social interaction and feeding behaviors, strengthening the idea that *Ssdp* plays a key role in the autism-associated phenotypes and is a major player in the 1p32.3 chromosomal region. We also observed differential expression in many immunity-related genes; however, these genes are predominantly expressed in peripheral tissues (fat bodies, trachea, and gut epithelia). Our RNA-seq analysis was performed with RNA isolated from fly heads; hence, it is possible that immunity-related transcripts might have originated from the fat body and/or neuronal cells.

Our RNA-seq data from the heads, highlighted the dysregulation of various genes involved in the oxidoreductase pathway, which was further validated in the brains. Our finding of reduced levels of *Dhrs4*, *CG40486*, and *catalase* mRNA in heads and brains of Ssdp[2082-G4] could be related to defective ROS clearance, as these genes are implicated in antioxidant function [127,128]. We also observed increased ROS and altered fission/fusion machinery in Ssdp[2082-G4] flies suggesting abnormal mitochondrial dynamics and function. These findings agree with the reported defects in mitochondria function and ROS levels in other NDDs, including autism [129–131].

Anxiety disorders are highly prevalent among individuals with autism, affecting nearly 40% of autistic individuals [132]. Similar to various rodent models and the TRPC6-based *Drosophila* model of ASD [20,133], *Ssdp* overexpressing flies, in our study also exhibited elevated anxiety in an open field arena. Furthermore, *Ssdp* knockdown and overexpressing flies showed enhanced reactivity to the repeated blue light stimuli compared to the controls, indicating a defective habituation process. The heightened sensory response to sudden and unexpected visual stimuli in *Ssdp* altered flies may be related to a cognitive intolerance of uncertainty, required to handle ambiguous and uncertain situations, which is known to be associated with anxiety [134].

Deficit in social interaction and communication skills are core features of ASD. Several animal models of ASD have recapitulated the abnormal social behavior observed in ASD individuals [80,135,136]. In our study, both *Ssdp* knockdown and overexpressing flies show impaired social behavior. Our *Drosophila* Ssdp model provides a valuable tool to investigate whether

*Ssdp* manipulations contribute to ASD-like phenotypes through defective synaptic development, synaptic transmission, or imbalance in excitatory/inhibitory synapses in the brain [21]. Furthermore, we also observed a deficit in multiple parameters of feeding in *Ssdp*-altered flies. This behavioral feature is consistent with the reported behavior in autistic children [137] and mouse models of autism [138].

Dosage sensitivity has been proposed as a major contributing factor in CNVs-mediated pathogenesis [11]. While, a significant proportion of gene dosage alterations in the genome have deleterious effects and are associated with diseases [139–142], very few genes are known to become pathological when their expression level enhances, potentially through promiscuous protein interactions [143]. Our data shows that *Ssdp* is one such dosage-sensitive gene, as both reduction and enhancement of *Ssdp* level had deleterious effects on cephalization, glial number, and autism-associated behaviors. Although, we observed rescue of defects in anxiety and habituation learning when elevated *Ssdp* mRNA levels in Ssdp[2082-G4] flies were normalized using Ssdp-RNAi, we did not observe any rescue in defects in social interaction. Suggesting that these behaviors are possibly governed by cells, which are extremely sensitive to alterations in *Ssdp* dosage. Furthermore, we show that pan-neuronal *Ssdp* knockdown exclusively in brains of adult flies was not sufficient to produce behavioral and functional defects. Altogether, these findings further support the role of *Ssdp* in neurodevelopment presented in this study.

Optogenetic neuronal activity and inactivity increased and decreased locomotion and social interactions, respectively, suggesting that Ssdp-expressing neurons are sufficient and necessary to regulate these 2 behaviors. However, anxiety-like wall-following behavior was decreased when Ssdp-expressing neurons were either optogenetically activated or silenced, while other behaviors such as stimulus reactivity and habituation remained defective, suggesting that these behavioral aspects are controlled by neurons independent from locomotion- and social interaction-regulating neurons.

It is important to consider the limitations of our study while interpreting our findings. Firstly, we have only focused on male flies to investigate the neurodevelopmental, functional, and behavioral consequences of *Ssdp* knockdown and overexpression. However, given that 1p32.3 can affect both males and females, and *Drosophila* exhibits sexual dimorphism in various behaviors related to autism [144,145], it is important to determine if *Ssdp* manipulations have similar or different behavioral and functional implications in female flies to fully understand the sex-specific effects of *Ssdp*.

Secondly, our study utilized only 1 transgene line each for overexpression and knockdown of *Ssdp* to interrogate its effects. Additional transgenes lines representing different levels of *Ssdp* gene expression could provide further insight into dose-dependent effects of *Ssdp* on neurodevelopment, function, and behavior.

Thirdly, our investigation of neuropil volume and glial number deficits was limited to third instar larvae and adult brains. Considering the neurodevelopment nature of ASD, it is crucial to investigate developmental and functional impairments across various stages of development to comprehensively understand the role of *Ssdp* in autism-associated phenotypes. Furthermore, we have shown partial rescue of behavioral deficits by normalized *Ssdp* levels or using optogenetic manipulations of Ssdp-expressing neuronal activity; however, we did not demonstrate the role played by identified *Ssdp* targets in this study, such as armadillo and catalase. Further studies targeting these specific pathways could provide additional mechanistic insight into the role of *Ssdp* in regulating autism-associated phenotypes.

In summary, despite these limitations, using a combination of molecular biology, genetics, and behavioral neuroscience in a *Drosophila melanogaster* model organism, our study provides evidence for a specific role of *Ssdp* in the development of the brain and autism-like behaviors. Although our study provides clues towards possible mechanisms through which *Ssdp* regulates

autism-associated phenotypes, such as via regulating gliogenesis, Wnt signaling, and oxidative reductase and mitochondrial function, the causal relationship is yet to be proven. Future work holds dissecting these pathways further through detailed analysis of the various components as well as attempting diverse rescue strategies. Further, research addressing these limitations will contribute to a better understanding of the pathophysiology of ASD-associated genes in *Drosophila* and for developing evidence-based therapies to reverse autism symptoms.

## Methods and materials

### Transgenic fly preparation and maintenance

All *Drosophila* fly strains used for the experiments were reared on Nutri-Fly MF food media (Genesee Scientific). They were maintained in 25˚C incubators with 12 h:12 h light-dark cycles and 70% humidity. Homozygous Ssdp[2084-G4] insertion is 100% lethal, thus heterozygous male progeny from Ssdp[2084-G4]/TM6b, Tb' (BDSC#65717) [34] crossed with w[1118] (BDSC#6326) were used as experiments and w[1118] flies were used as controls. For brain-specific pan-neuronal knockdown, Elav-Gal4/cyo (BDSC#8765) driver was crossed with UAS-Ssdp-RNAi (BDSC#62167), and each line crossed with w[1118] served as genotypic controls. 6xUAS-GFP (BDSC#52262) was used to check the expression pattern of Ssdp[2084-G4]-Gal4. For *Ssdp* knockdown in Ssdp-expressing cells, Ssdp-Gal4 (i.e., Ssdp[2084-G4]/TM6b, Tb') flies were crossed with UAS-Ssdp-RNAi flies. To knockdown *Ssdp* pan-neuronally exclusively in adult flies, crosses between Elav-Gal4/cyo, Tub-Gal80[ts]/TM2 and UAS-Ssdp-RNAi, and w[1118] and Elav-Gal4/cyo, Tub-Gal80[ts]/TM2 were maintained at 21˚C. After emergence, the F1 progeny were selected against balancers and conditioned at 31˚C for 48 h and were transferred to 25˚C overnight, following which experiments were conducted at 25˚C. All experiments were performed using male flies aged 3 to 4 days.

### Dm-Ssdp and Hs-SSBP3 sequence alignment

The protein sequences of DM-SSDP and HS-SSBP3 were obtained from the UniProtKB database (accession numbers Q8MZ49 and Q9BWW4, respectively) and saved in FASTA format as described previously [146]. Clustal-Omega 1.2.4 software was employed to align the sequences with default parameters. Conserved regions, percentage identity, and percentage similarity were analyzed from the aligned sequences to compare the 2 sequences. Based on the results of the sequence alignment, conclusions were drawn regarding the degree of similarity between DM-SSDP and HS-SSBP3.

### Scanning electron microscopy

We performed high-resolution SEM imaging of fly body parts (head, wings, and thorax). Fly samples were affixed to SEM aluminum stubs using conductive adhesive carbon tape or carbon glue. To enhance electrical conductivity and imaging quality, the samples were coated with a 10-nm layer of gold using a Quoram 150T ES magnetron sputter coater (Quoram Technology) with argon gas. Imaging was performed using a FEI Quanta650FEG SEM (Thermo Scientific, United States of America) operating at 5 kV with a Secondary Electron ETD detector.

### Immunohistochemistry and microscopy

Adult brains and CNS or wing discs from third instar larvae were dissected in chilled 1× PBS and fixed with 4% paraformaldehyde for 20 min. Samples were washed thrice with 1× PBST (1× PBS with 1% Triton X-100) for 15 min. Primary antibodies: Mouse anti-DLG (DSHB:

4F3) 1:50, Mouse anti-Bruchpilot (nc82, DSHB), Mouse anti-Repo (DSHB: 8D12), Mouse anti-Elav (DSHB: 9F8A9). Secondary antibodies: Anti-mouse-568 (Thermo Fisher Scientific A11031) 1:200, Anti-Armadillo (DSHB: N27A1) 1:200, Anti-Wingless (DSHB: 4D4) 1:100. The samples were incubated with respective primary antibody overnight at 4˚C and washed thrice using 1× PBST for 15 min. The samples were then incubated with appropriate secondary antibody for 3 h at room temperature and washed thrice using 1× PBST for 15 min. For anti-nc82 staining, primary antibody was incubated for 24 h, and secondary antibody was incubated overnight at 4˚C. Brains, CNS, or wing discs were mounted on a glass slide using Vecta-shield (Vectorlabs) with a coverslip. The samples were scanned using Nikon A1R Multiphoton Confocal Microscope (2 μm z-stacks).

## Quantitative real-time PCR

Twenty fly heads or brains were homogenized in TRI Reagent (Sigma) to isolate RNA. Approximately 2,000 ng RNA was converted to cDNA using the High-Capacity cDNA Reverse Transcription Kit (Applied Biosystems). qRT-PCR was performed using QuantStudio (Applied Biosystem) using PowerUp SYBR Green Master Mix (Thermo Fisher) and primers (Table 1). Analysis was performed using the $2^{-\Delta\Delta CT}$ method. Total cDNA concentration was normalized to endogenous *Rpl32* expression. Data are represented as mean fold change relative to controls.

## RNA sequencing and analysis

The RNA sequencing and DEG analysis was performed at MacroGen (www.macrogen.com, Seoul, Korea). Total RNA was extracted as described above in the qRT-PCR section from heads of 3- to 4-day-old males of w[1118] and Ssdp[2082-G4]/+ flies. The RNA was quantified first

**Table 1. List of primers used in the qRT-PCR analysis.**

| Gene name | Sequence |
| --- | --- |
| *Ssdp* | For: TACGGCAAATCAAAGACATCGG<br>Rev: CAGATATTCGTACACGTACAGGG |
| *Catalase* | For: TTCTGGTTATCCCGTTGAGC<br>Rev: GGTAATGGCACCAGGAGAAA |
| *Sepia* | For: CACAGAGTCGCTACTGATTTGT<br>Rev: CGCTCGATTAGTAACTTGTCCTG |
| *GstE1* | For: TCTTCTTCGATGCCAGTGTAATC<br>Rev: TCGGTAACACCGTTTATCCAAAA |
| *CG40486* | For: CTGAACCCCGGACAACTTCTC<br>Rev: GCTGCATTGAGCGAAAGGC |
| *Gdap1* | For: TGCAGGACTTCAAGGCTCC<br>Rev: CACAGGTCCACCACATAGGG |
| *Dhrs4* | For: GGCCCTAATCTCAATCAATGCC<br>Rev: GGCTACTTTTCCTGCCAAACG |
| *Drp1* | For: CTCCGGCAAGAGTTCGGTG<br>Rev: GGCGACGGGTCACAATACC |
| *Fis1* | For: GTCTGGCTTAAAATACTGCCGA<br>Rev: CATACCCTTTGCCACTTCCTT |
| *Marf* | For: ATGGCGGCCTACTTGAACC<br>Rev: GCGATGAGTTGAACCGGGA |
| *Opa1* | For: TCAAGCTGCGATACATCGTCC<br>Rev: GGCAGTCCATCCTTCCATTCC |
| *Rpl32* | For: CGGATCGATATGCTAAGCTGT<br>Rev: GCGCTTGTTCGATCCGTA |

using NanoDrop and then using highly sensitive Qubit 3.0. Further, RNA integrity was checked on high sensitivity tape (Agilent TapeStation 2200). Samples with RIN number >8.5 were selected for library construction. Three w[1118] and 3 Ssdp[2082-G4]/+ samples were used to prepare a 100 bp paired sequencing library using the TruSeq Stranded mRNA Library prep kit. Distribution of PCR-enriched fragments were checked using Agilent Technologies 2100 Bioanalyzer using a DNA 1000 chip and >40 M read per sample were sequenced on an Illumina NovaSeq6000. The dm6 was used as a reference to the *Drosophila* genome. Bowtie2 aligner was used to paired-end reads to the reference transcriptome. Known genes and transcripts were assembled with StringTie based. After assembly, the abundance of gene/transcript was calculated in the read count and normalized values as FPKM (Fragments Per Kilobase of transcript per Million mapped reads) and TPM (Transcripts Per Kilobase Million) for a sample. Low-quality transcripts were filtered. The genes with zero count in any sample were removed. Therefore, from a total of 17,829 genes, 5,982 were excluded and only 11,847 genes were used for statistical analysis. Afterwards, RLE normalization was performed. The read count data was normalized with the relative log expression (RLE) method in DESeq2 R library. The DEG analysis was performed on a comparison pair (SSDP_vs_CTL) using DESeq2, genes were selected if they satisfied |fc|> = 1.5 and nbinomWaldTest raw $p$-value <0.05 conditions in a comparison pair (3 SSDP_vs_ 3 CTL). RNA-seq data are available at the Gene Expression Omnibus repository GEO:GSE220311.

## Brain volume measurement and counting glial cells

To determine the central brain volume, brains were stained with anti-DLG or anti-nc82 antibodies. The volume was measured using the 3D Manager plugin in Fiji. Briefly, the optic lobes were cleared across all the stacks, and a Gaussian blur filter (Sigma = 2) was applied. A threshold was applied to all the stacks to select the central brain region and the 3D Manager plugin was employed to obtain the volume in microns.

To determine the number of glial cells, brains were stained with anti-Repo antibody. The number of glial cells across all stacks was measured using the 3D Objects Counter feature in Fiji. Firstly, the noise was filtered out by applying a difference of Gaussians between the original stacks (Sigma = 1) and duplicate stacks (Sigma = 2). Background staining was further reduced using the Subtract Background feature. The 3D Objects Counter feature was then used to apply a threshold such that the glial cells were selected across all stacks, and the total count of glial cells was inferred from the resultant number of objects detected.

## Synaptic density measurement

Synaptic density measurement was performed as described in [146]. Brains were stained with anti-nc82 antibodies and imaged at 100× oil immersion (Nikon A1R Multiphoton Confocal Microscope). The SEZ region and both SLP regions were scanned separately for each brain. Fiji 3D Object Counter plugin was used to measure the synaptic contacts through 39 stacks in the SEZ region and 41 stacks in the SLP region. Same threshold [146] was applied across all the samples to return the total number of contiguous voxel elements across the images. The number of objects detected was plotted.

## Western blot

Thirty fly heads were lysed in triton lysis buffer (50 mM Tris-HCl (pH 7.4), 1% Triton X-100, 150 mM NaCl, 1 mM EDTA, 1× protease inhibitor, and 1 mM PMSF) and homogenized in a Fisher Scientific Bead Mill 4. The triton-soluble fractions were run on a 10% SDS-PAGE gel and transferred onto a nitrocellulose membrane. Primary antibodies used: Mouse anti-

Armadillo (DSHB: N27A1) L, Mouse anti-Wingless (DSHB: 4D4), Mouse anti-Tubulin (Sigma: T6074). Secondary antibody used: Anti-mouse-HRP (Thermo Fisher Scientific) 1:10,000.

### $H_2$DCFDA fluorescence analysis

To determine the levels of ROS in the brains, we used 10 mM $H_2$DCFDA (Invitrogen D399) as previously described [147]. An oval of dimensions 240.31 × 132.58 μm was used to measure fluorescence signals in the SLP and SEZ regions. The background mean intensity values were subtracted from the signal mean intensity values. For the SLP region, analysis was performed per hemisphere.

### Mitochondrial morphology analysis

To assess the morphology of mitochondria in the SEZ and SLP regions of Ssdp[2082-G4]-Gal4 and Elav-Gal4 controls driving UAS-Mito-Ds-Red (BDSC#93056), MIP were generated from selected z-stacks for each brain scanned at 100× using a spinning disk microscope (CSU-X1A, Nikon). Two SEZ and SLP regions were selected in experimental and control brains. The data from both hemispheres were averaged to obtain the measurements per brain. For area and circularity measurements, area and shape descriptors were selected under set measurements, and particles were analyzed. For length, the thresholded image was skeletonized, particles were analyzed, and length was inferred from the area of the skeletons.

### Open field assay

To assess the locomotor and exploratory behavior of the *Drosophila* flies, we used an open field assay in multiple 15 mm circular arenas. The fly motor activity was recorded, tracked, and analyzed using DART [69]. *Drosophila* flies were anesthetized on ice and loaded into each arena. Flies were incubated at 25˚C for 15 to 20 min before behavioral recording. The trajectory was recorded and tracked for 10 min at 15 frames per second (fps).

### Visual startle and habituation assay

To assess nonassociative learning, we modified the previously published light-switch off assay [71]. We used a blue light-on after-dark period paradigm in which a blue light of maximum intensity was shone 5 times for one-second at one-minute intervals. We used DART to track flies' movement in a 65 mm × 5 mm × 2 mm rectangular arena at 15 fps. Before each experiment, flies were habituated in the dark for 15 to 20 min at 25˚C. Each fly's baseline activity was measured using locomotion traces as an active average speed during the first 40 s. Their startle response to blue light was assessed using an active average speed of 10 s before and after the first pulse and their habituation as the locomotor reactivity to the fifth pulse.

### Social interaction analysis

Four flies were placed in each circular arena (35 mm diameter), and their movements were recorded using the OnePlus 9 Android smartphone (50 megapixels) at 15 fps for 20 min at 25˚C. Prior to recording, the flies were habituated in the arena for 20 min at 25˚C. Using Iowa-FLI Tracker [148], locomotor tracks of each of the 4 flies were computed, outputting social interaction parameters like interaction time and the number of contacts. Sitting interactions were used for analysis. Nonmoving flies were excluded from the analysis.

## Feeding assay

To assess the feeding parameters of the flies, the FlyPAD system was used [87]. Flies were starved overnight (16 to 20 h) on 0.8% agarose at 21˚C. The following day, individual flies were placed in the FlyPAD arenas, loaded with melted 25 mM sucrose in 0.8% agarose. The feeding behavior was recorded for 1 h at 25˚C. Data analysis was performed using FlyPAD software in Matlab (Mathworks) [63,87].

## Optogenetic experiments

F1 male progeny aged 0 to 2 days from Ssdp[2084-G4]/TM6b, Tb' crossed with 20X-UAS-CsChrimson (BDSC #55134) or 20X-UAS-GtACR1 [91] were reared on media mixed with 1 mM all-trans-retinal (ATR, Carbosynth, #16-31-4). ATR was prepared in absolute ethanol. The F1 progeny of 20× UAS-CsChrimson and UAS-GtACR1 lines crossed with w[1118] served as genotypic controls. All behavioral experiments were performed after at least 48 to 72 h of retinal treatment. All the steps were performed with minimal light exposure. The red (635 nm) and green (532 nm) light intensities were measured using the optical power meter (Thorlabs PM400) and optical sensor (Thorlabs S120C).

## Statistics

Error bars in all data figures are 95% confidence intervals for the difference in means. All data points are presented as means unless otherwise stated. Shapiro–Wilk normality test was used to test whether the data followed a normal distribution. The statistical testing was combined with calculating the effect size. Cohen's d was used as the effect size measure for data with normal distribution, and for nonparametric distributed data, effect sizes were calculated using Cliff's Δ. Effect size analysis and significance tests were conducted using estimationstats.com [149]. For qRT-PCR experiments, $p$-values were calculated using Student's $t$ test.

## Supporting information

**S1 Fig. Gene map depicting Ssdp[2082-G4] transgenic insertion.** Mapped positions show presence of H.M.S. Beagle, a natural transposable elements (TE) as well as other TEs upstream of the Ssdp[2082-G4]-Gal4 insertion, which may enhance the transcription of nearby loci. Created with BioRender.com.
(TIF)

**S2 Fig. Ssdp-positive cells colocalize with neurons and not with glia in Ssdp[2082-G4] brains.** (A and B) 100× magnification images showing expression pattern of 6x-UAS-GFP driven by Ssdp[2082-G4]-Gal4 in the SLP region with anti-Elav and anti-Repo staining, respectively. White arrows point towards co-localization with Elav-positive neurons in yellow. Scale bar = 10 μm.
(TIF)

**S3 Fig. KEGG pathways analysis for the down-regulated genes in *Ssdp* expressing heads showing mapping of genes related to glutathione metabolism.**
(TIF)

**S4 Fig. KEGG pathways analysis for the down-regulated genes in *Ssdp* expressing heads showing mapping of genes related to Toll and Imd signaling pathway.**
(TIF)

**S5 Fig. KEGG pathways analysis for the up-regulated genes in *Ssdp* expressing heads showing mapping of genes related to FoxO signaling pathway.**
(TIF)

**S6 Fig. KEGG pathways analysis for the down-regulated genes in *Ssdp* expressing heads showing mapping of genes related to Wnt signaling.**
(TIF)

**S7 Fig. Validation of mitochondrial fission/fusion genes in heads of Ssdp overexpressing flies.** Both *Drp1* ($p = 0.0095$, $n = 5$) and *Marf1* ($p = 0.045$, $n = 5$) levels are significantly up-regulated in the brains of Ssdp[2082-G4]/+ flies compared to w[1118] controls. *P*-values were obtained by Student's *t* test. For all qRT-PCR experiments, expression was normalized to *Rpl32*. Horizontal red line represents the mean value. Each black dot is a data point representing an independent biological replicate with 20 individual fly heads per biological replicate. The raw data underlying this figure can be found in S1 Data file.
(TIF)

**S8 Fig. *Ssdp* knockdown and overexpression flies show multiple autism-like phenotypes.** (A) Heatmap occupancy plots of wall-following behavior in an open field were averaged from the recorded behavior of all flies in each group; blue indicates the lowest occupancy in that region, and red indicates maximum occupancy. (B) Elav-Gal4>UAS-Ssdp-RNAi exhibit a heightened baseline locomotor speed ($n = 38–77$, Cohen's d = 0.53 [95 CI 0.102, 0.985], $p = 0.008$) and startle to the first blue pulse than w[1118]>Elav-Gal4 and w[1118]-Ssdp-RNAi controls. (C) Heterozygous Ssdp[2082]/+ flies also show increased baseline locomotor speed ($n = 40$, Cohen's d = 0.72 [95 CI 0.298, 1.09], $p = 0.001$) and startle to the first blue pulse compared to w[1118] controls. (D) History of sitting fly interactions represents an interactogram for 20 min for each genotype. Each vertical line's thickness represents time spent interacting with another fly (left). Number of direct social contacts between flies in a group are reduced in both Elav-Gal4>UAS-Ssdp-RNAi ($n = 36–41$, Cliff's Δ = −0.35 [95 CI −0.55, −0.13], $p = 0.0018$) and Ssdp[2082]/+ flies ($n = 46–48$, Cliff's Δ = −0.22 [95 CI −0.44, 0.016], $p = 0.067$) compared to controls (right); however, for Ssdp[2082]/+ flies the decrease is not statistically significant. (E) Histogram depicts reduced inter-sip intervals in Elav-Gal4>UAS-Ssdp-RNAi and Ssdp[2082]/+ flies ($n = 35–45$) (left). Line plots show cumulative sips over the assay time, which are decreased in Elav-Gal4>UAS-Ssdp-RNAi and Ssdp[2082]/+ flies compared to their respective controls (1 h, $n = 35–45$) (middle). The schematic on the right shows the architecture of the feeding behavior of genotypic control and experimental flies (not drawn to scale). In scatter plots, each dot represents the mean value for a single fly. The red horizontal line represents the mean value. *P*-values are from the permutation *t* test. The raw data underlying this figure can be found in S1 Data file.
(TIF)

**S9 Fig. *Ssdp* knockdown in Ssdp-Gal4 labeled cells partially rescues autism-associated behaviors.** (A) Relative *Ssdp* mRNA expression in the heads of heterozygous Ssdp[2082]/+ flies was enhanced ($p = 0.049$, $n = 10–13$) compared to w[1118] controls, but was not different in heads of Ssdp[2082-G4]>UAS-Ssdp-RNAi flies compared to w[1118] controls ($p = 0.26$, $n = 5–10$) and Ssdp[2082]/+ flies ($p = 0.16$, $n = 5–13$). Each black dot is a data point representing an independent biological replicate with 20 individual fly heads per biological replicate. Total cDNA concentration was normalized to endogenous *Rpl32* expression. *P*-values are calculated using the Student's *t* test. (B) The active average speed of Ssdp[2082-G4]/+ flies was similar to control flies ($n = 46–47$, Cliff's Δ = 0.0014 [95 CI −0.235, 0.24], $p = 0.99$); however, Ssdp[2082-G4]>UAS-Ssdp-RNAi flies moved slower in open field compared to w[1118] controls ($n = 46–49$,

Cliff's Δ = −0.48 [95 CI −0.667, −0.248], $p < 0.0001$) and Ssdp[2082]/+ flies ($n = 47$–49, Cliff's Δ = −0.59 [95 CI −0.755, −0.391], $p < 0.0001$). (C) Heterozygous Ssdp[2082-G4]/+ flies show a significantly higher wall-following compared to controls ($n = 49$–57, Cliff's Δ = 0.47 [95 CI 0.256, 0.651], $p < 0.0001$); however, Ssdp[2082-G4]>UAS-Ssdp-RNAi flies show wall-following behavior similar to controls ($n = 57$–61, Cliff's Δ = −0.031 [95 CI −0.246, 0.183], $p = 0.78$) and significantly decreased wall-following compared to Ssdp[2082]/+ flies ($n = 49$–61, Cliff's Δ = −0.62 [95 CI −0.768, −0.428], $p < 0.0001$). (D) Ssdp[2082-G4] flies show defective habituation speed to the fifth pulse ($n = 44$–47, Cohen's d = 0.82 [95 CI 0.391, 1.27], $p < 0.0001$), Ssdp[2082-G4]>UAS-Ssdp-RNAi flies show no habituation defect compared to w[1118] controls ($n = 44$–46, Cohen's d = 0.39 [95 CI −0.0448, 0.796], $p = 0.064$) and a significantly decreased habituation speed compared to Ssdp[2082]/+ flies ($n = 46$–47, Cohen's d = −0.49 [95 CI −0.923, −0.0548], $p = 0.02$. (E) Both Ssdp[2082-G4]/+ ($n = 44$, Cliff's Δ = −0.43 [95 CI −0.63, −0.185], $p = 4.0 \times 10^{-04}$) and Ssdp[2082-G4]>UAS-Ssdp-RNAi ($n = 44$, Cliff's Δ = −0.50 [95 CI −0.688, −0.276], $p < 0.0001$) flies show reduced interaction time compared to w[1118] controls. No significant difference was observed in the interaction time between Ssdp[2082-G4]/+ and Ssdp[2082-G4]>UAS-Ssdp-RNAi flies ($n = 44$, Cliff's Δ = −0.11 [95 CI −0.349, 0.135], $p = 0.36$). Both Ssdp[2082-G4]/+ ($n = 44$, Cliff's Δ = −0.41 [95 CI −0.602, −0.171], $p = 2.0 \times 10^{-04}$) and Ssdp[2082-G4]>UAS-Ssdp-RNAi ($n = 44$, Cliff's Δ = −0.29 [95 CI −0.512, −0.0501], $p = 0.018$) flies made fewer social contacts compared to w[1118] controls. No significant difference was observed in the number of contacts between Ssdp[2082-G4]/+ and Ssdp[2082-G4]>UAS-Ssdp-RNAi flies ($n = 44$, Cliff's Δ = 0.088 [95 CI −0.162, 0.332], $p = 0.48$). In scatter plots, each dot represents the mean value for a single fly. Horizontal red line represents the mean value. *P*-values are from a two-sided permutation *t* test. The raw data underlying this figure can be found in S1 Data file.

(TIF)

**S10 Fig.** *Ssdp* **knockdown exclusively in adult flies does not produce behavioral and functional defects.** (A) Schematic depicting the timeline of experimental procedure on w[1118]>Elav-Gal4, Tub-Gal80[ts] and Elav-Gal4, Tub-Gal80[ts]>UAS-Ssdp-RNAi flies. Crosses were maintained at 21°C, after eclosion the adult F1s were conditioned at 31°C for 48 h, then kept at 25°C overnight, following which behavioral and functional assays were performed. Created with [BioRender.com](BioRender.com). (B) In the open field arena, no changes were observed in average speed ($n = 32$–40, Cliff's Δ = −0.04 [95 CI −0.32, 0.23], $p = 0.79$) and percentage time spent in outer edge ($n = 29$–30, Cliff's Δ = 0.04 [95 CI −0.27, 0.34], $p = 0.78$); however, wall approach angles were statistically significantly different between Elav-Gal4, Tub-Gal80[ts]>UAS-Ssdp-RNAi flies and w[1118]>Elav-Gal4, Tub-Gal80[ts] controls. Fly representations in wall approach angles graph created with [BioRender.com](BioRender.com). (C) Habituation speed to the fifth blue light pulse is unaltered in Elav-Gal4, Tub-Gal80[ts]>UAS-Ssdp-RNAi flies compared to w[1118]>Elav-Gal4, Tub-Gal80[ts] controls ($n = 30$–40, Cliff's Δ = 0.22, [95 CI −0.0625, 0.47], $p = 0.12$). (D) In the social interaction assay, no differences were observed in the interaction time ($n = 28$–40, Cliff's Δ = 0.15 [95 CI −0.15, 0.42], $p = 0.31$) and number of contacts ($n = 28$–40, Cliff's Δ = 0.053 [95 CI −0.22, 0.32], $p = 0.72$) between Elav-Gal4, Tub-Gal80[ts]>UAS-Ssdp-RNAi flies and w[1118]>Elav-Gal4, Tub-Gal80[ts] controls. (E) Pseudocolor representative images of w[1118]>Elav-Gal4, Tub-Gal80[ts] and Elav-Gal4, Tub-Gal80[ts]>UAS-Ssdp-RNAi brains stained with H$_2$DCFDA. White dashed ovals mark regions of interest in the SLP and SEZ. H$_2$DCFDA fluorescence is not different in both SLP ($n = 10$, Cohen's d = −0.063 [95 CI −1.03, 0.919], $p = 0.89$) and SEZ ($n = 5$, Cohen's d = 0.23 [95 CI −1.25, 1.82], $p = 0.72$) regions between Elav-Gal4, Tub-Gal80[ts]>UAS-Ssdp-RNAi flies and w[1118]>Elav-Gal4, Tub-Gal80[ts] controls. In scatter plots, each dot represents the mean value for a single fly. Horizontal red line represents the

mean value. *P*-values are from a two-sided permutation *t* test. The raw data underlying this figure can be found in S1 Data file.
(TIF)

**S11 Fig. Alteration of autism-associated behaviors upon optogenetic manipulations.** (A) Startle speed is significantly increased in heterozygous Ssdp[2082-G4]/+ flies (w[1118], *n* = 43, Cohen's d = −0.12 [95 CI −0.312, 0.0533], *p* = 0.183; Ssdp[2082-G4]/+, *n* = 43, Cohen's d = 0.21 [95 CI 0.105, 0.359], *p* = $6 \times 10^{-04}$); however, it is significantly decreased in Ssdp[2082-G4]>UAS-Chrimson (w[1118]>UAS-Chrimson, *n* = 43, Cohen's d = −0.26 [95 CI −0.544, 0.0314], *p* = 0.083; Ssdp[2082-G4]>UAS-Chrimson, *n* = 43, Cohen's d = −0.385 [95 CI −0.674, −0.146], *p* = 0.0044) and Ssdp[2082-G4]>UAS-GtACR1 (w[1118]>UAS-GtACR1, *n* = 38, Cohen's d = 0.00153 [95 CI −0.258, 0.239], *p* = 0.99; Ssdp[2082-G4] >UAS-GtACR1, *n* = 34, Cohen's d = −0.32 [95 CI −0.605, −0.0846], *p* = 0.0064) flies compared to their controls. Schematic created with BioRender.com. (B) Number of contacts is significantly decreased in Ssdp[2082-G4]/+ flies (*n* = 44–47, Cliff's Δ = −0.48 [95 CI −0.662, −0.26], *p* < 0.0001) and Ssdp[2082-G4] >UAS-GtACR1 flies (*n* = 35–42, Cliff's Δ = −0.26 [95 CI −0.478, −0.0285], *p* = 0.026); however, it is significantly increased in Ssdp[2082-G4]>UAS-Chrimson flies (*n* = 35–42, Cliff's Δ = 0.31 [95 CI 0.0374, 0.546], *p* = 0.018) compared to their controls. Schematic created with BioRender.com. In scatter plots, each dot represents the mean value for a single fly. Horizontal red line represents the mean value. *P*-values are from a two-sided permutation *t* test. The raw data underlying this figure can be found in S1 Data file.
(TIF)

**S1 Table. Table depicting the genes present in 1p32.3 chromosomal region.** Human genes that have orthologs in *Drosophila melanogaster* and their DIOPT scores are presented. The stock numbers for available transgenic lines in Bloomington *Drosophila* Stock Center (BDSC) or Vienna *Drosophila* Resource Center (VDRC) are mentioned.
(DOCX)

**S2 Table. Dysregulated genes in *Ssdp* overexpressing flies as identified by RNA-seq.** A fold change of 1.5 was set as the criteria, based on the fold change of *Ssdp* gene. According to this criteria, 256 genes were down-regulated and 160 genes were up-regulated. The fold change, *p*-value, human ortholog, and DIOPT score are mentioned for each dysregulated gene.
(DOCX)

**S3 Table. Association of down-regulated genes in *Ssdp* overexpressing flies with neurodevelopmental defects.** Down-regulated genes in *Ssdp* overexpressing flies identified using RNA-seq, with an orthology score ≥5 are presented. The human ortholog, fold change, DIOPT score for these genes are mentioned. Furthermore, the number of patients listed in DECIPHER with variations in these genes that show autistic behavior, micro/macrocephaly, and ID are presented. Evidence of association of these genes with autism, ID, other neurological disorders, and inflammation/immunity response in literature is also listed.
(DOCX)

**S4 Table. Association of up-regulated genes in *Ssdp* overexpressing flies with neurodevelopmental defects.** Up-regulated genes in *Ssdp* overexpressing flies identified using RNA-seq, with an orthology score ≥5 are presented. The human ortholog, fold change, DIOPT score for these genes are mentioned. Furthermore, the number of patients listed in DECIPHER with variations in these genes that show autistic behavior, micro/macrocephaly, and ID are presented. Evidence of association of these genes with autism, ID, other neurological disorders,

and inflammation/immunity response in literature is also listed.
(DOCX)

**S1 Data. Raw data for each figure in the manuscript.** Each sheet in the Excel file corresponds to a specific figure panel.
(XLSX)

**S1 Raw Image. The original blots for Fig 6A stained with anti-Armadillo, anti-Wingless, or anti-Tubulin antibodies.** Lanes labeled with X were not depicted in the blot in Fig 6A but have been used to perform quantification depicted in the bar graphs.
(PDF)

**S1 Movie. The Z-stacks of w[1118] larval CNS stained with anti-DLG antibody.**
(AVI)

**S2 Movie. The Z-stacks of w[1118]-Ssdp[2082 = G4] larval CNS stained with anti-DLG antibody.**
(AVI)

**S3 Movie. The Z-stacks of w[1118] adult brain stained with anti-DLG antibody.**
(AVI)

**S4 Movie. The Z-stacks of w[1118]-Ssdp[2082 = G4] adult brain stained with anti-DLG antibody.**
(AVI)

**S5 Movie. The Z-stacks of w[1118] adult brain stained with anti-nc82 antibody.**
(AVI)

**S6 Movie. The Z-stacks of w[1118]-Ssdp[2082 = G4] adult brain stained with anti-nc82 antibody.**
(AVI)

**S7 Movie. The Z-stacks of objects map generated through Fiji upon 3D glial number counting of w[1118] larval CNS stained with anti-Repo antibody.**
(AVI)

**S8 Movie. The Z-stacks of objects map generated through Fiji upon 3D glial number counting of w[1118]-Ssdp[2082 = G4] larval CNS stained with anti-Repo antibody.**
(AVI)

**S9 Movie. The Z-stacks of objects map generated through Fiji upon 3D glial number counting of w[1118] adult brain stained with anti-Repo antibody.**
(AVI)

**S10 Movie. The Z-stacks of objects map generated through Fiji upon 3D glial number counting of Ssdp[2082 = G4]-EGFP adult brain stained with anti-Repo antibody.**
(AVI)

**S11 Movie. The Z-stacks of w[1118]-Elav larval CNS stained with anti-DLG antibody.**
(AVI)

**S12 Movie. The Z-stacks of Elav-Ssdp-RNAi larval CNS stained with anti-DLG antibody.**
(AVI)

**S13 Movie. The Z-stacks of w[1118]-Elav adult brain stained with anti-DLG antibody.**
(AVI)

**S14 Movie. The Z-stacks of Elav-Ssdp-RNAi adult brain stained with anti-DLG antibody.**
(AVI)

**S15 Movie. The Z-stacks of objects map generated through Fiji upon 3D glial number counting of w<sup>1118</sup>-Elav larval CNS stained with anti-Repo antibody.**
(AVI)

**S16 Movie. The Z-stacks of objects map generated through Fiji upon 3D glial number counting of Elav-Ssdp-RNAi larval CNS stained with anti-Repo antibody.**
(AVI)

**S17 Movie. The Z-stacks of objects map generated through Fiji upon 3D glial number counting of w<sup>1118</sup>-Elav adult brain stained with anti-Repo antibody.**
(AVI)

**S18 Movie. The Z-stacks of objects map generated through Fiji upon 3D glial number counting of Elav-Ssdp-RNAi adult brain stained with anti-Repo antibody.**
(AVI)

## Acknowledgments

We thank Dr. Fadel Tissir, Dr. Omar Khan, Dr. Johan Ericsson, and Farhan lab members for their valuable comments and suggestions. Authors express their gratitude to Dr. Johanna M. Dela Cruz, BRC Imaging Facility, Institute of Biotechnology, Cornell University, USA, for her invaluable assistance in the methodology for 3D analysis of brain volume and glial count. Authors thank Mujaheed Pasha at HBKU Core Labs for helping in SEM imaging.

## Author Contributions

**Conceptualization:** Farhan Mohammad.

**Data curation:** Ayesha Banu, Swetha B. M. Gowda, Foysal Ahammad.

**Formal analysis:** Safa Salim, Sadam Hussain, Ayesha Banu, Foysal Ahammad, Farhan Mohammad.

**Funding acquisition:** Farhan Mohammad.

**Investigation:** Safa Salim, Sadam Hussain, Ayesha Banu, Swetha B. M. Gowda, Foysal Ahammad, Amira Alwa, Mujaheed Pasha, Farhan Mohammad.

**Methodology:** Farhan Mohammad.

**Project administration:** Farhan Mohammad.

**Resources:** Farhan Mohammad.

**Software:** Farhan Mohammad.

**Supervision:** Farhan Mohammad.

**Validation:** Safa Salim, Sadam Hussain, Amira Alwa, Mujaheed Pasha.

**Visualization:** Safa Salim, Foysal Ahammad, Farhan Mohammad.

**Writing – original draft:** Safa Salim, Farhan Mohammad.

**Writing – review & editing:** Farhan Mohammad.

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
