## [Editor Report · Decision Letter 0]

7 Dec 2022

Dear Dr Farhan, 

Thank you for adding this additional work to your manuscript entitled "Ssdp influences neurodevelopment and autism-like behaviors in Drosophila melanogaster" and for asking us to consider it further at PLOS Biology.

I looked over the additional evidence provided and also passed this by one of our academic editors with relevant expertise. I am writing now to let you know that we would like to send your submission out for external peer review. I do, however, want to note that our Academic Editor did have some concerns as to whether this would ultimately be a strong enough candidate for PLOS Biology, citing some concerns with how clearly the global changes you see are related to changes in Wnt signaling, and how the intracellular/molecular level changes relate to the cellular level (axons and dendrites) changes. With this in mind, we will be looking for strong reviewer support to move forward at PLOS Biology.

Before we can send your manuscript to reviewers, we need you to complete your submission by providing the metadata that is required for full assessment. To this end, please login to Editorial Manager where you will find the paper in the 'Submissions Needing Revisions' folder on your homepage. Please click 'Revise Submission' from the Action Links and complete all additional questions in the submission questionnaire.

Once your full submission is complete, your paper will undergo a series of checks in preparation for peer review. After your manuscript has passed the checks it will be sent out for review. To provide the metadata for your submission, please Login to Editorial Manager (https://www.editorialmanager.com/pbiology) within two working days, i.e. by Dec 09 2022 11:59PM.

Kind regards,

Kris

Kris Dickson, Ph.D., (she/her)

Neurosciences Senior Editor/Section Manager

PLOS Biology

kdickson@plos.org

---

## [Decision Letter · Decision Letter 1]

1 Mar 2023

Dear Dr Farhan,

Thank you again for your patience while your manuscript "Ssdp influences neurodevelopment and autism-like behaviors in Drosophila melanogaster" was peer-reviewed at PLOS Biology. It has now been evaluated by the PLOS Biology editors, an Academic Editor with relevant expertise, and by several independent reviewers. 

In light of the reviews, which you will find at the end of this email, we would like to invite you to revise the work to thoroughly address the reviewers' reports.

As you will see, the reviewers appreciate the importance of the topic and think the findings are potentially interesting, however they have raised a number of important and overlapping concerns which would need to be thoroughly addressed before we can consider your manuscript further for publication. After discussion with the Academic Editor, we think it would be particularly important to validate the tools (RNAi) and methods (image quantification) as the reviewers suggest, and to examine at least one developmental stage and show that there are developmental defects, and to identify their nature.

Given the extent of revision needed, we cannot make a decision about publication until we have seen the revised manuscript and your response to the reviewers' comments. Your revised manuscript is likely to be sent for further evaluation by all or a subset of the reviewers.

We expect to receive your revised manuscript within 3 months, however, as we think that a substantial amount of new data may be required to address the reviewer requests, we would be willing to grant an extension if needed. Please email us (plosbiology@plos.org) if you have any questions or concerns, or would like to request an extension. 

**IMPORTANT - SUBMITTING YOUR REVISION**

*Re-submission Checklist*

*Published Peer Review*

*PLOS Data Policy*

*Blot and Gel Data Policy*

Sincerely,

Lucas

Lucas Smith, Ph.D.

Associate Editor

PLOS Biology

lsmith@plos.org

REVIEWS:

Reviewer #1: 

This paper by Safa Salim et al. makes the following conclusions: that the human gene SSBP3 located in the chromosome band 1p32.3, whose microdeletion/duplication is associated with neurodevelopmental defects and autism has its Drosophila melanogaster ortholog called Ssdp. Through a knockdown and overexpression approaches they showed that Ssdp plays a role in the "brain development" and autism-like behaviors in flies. At the cellular and molecular levels, Ssdp regulates gliogenesis, Wnt signaling, and oxidative reductase and mitochondrial function. Understanding the mechanism of neurodevelopmental disorders (NDDs) such as autism caused by genomic dysfunctions including chromosomal rearrangements is important. Aberrations in 1p32.3 have been reported in patients with NDDs, but the link between genes located within the 1p32.3 region and autism spectrum disorders (ASD) remains unclear. However, there are a number of issues that would need to be addressed before a possible publication.

Major

1) The most significant major comment I have is related to the evidence on neuro-developmental defects or disorders in the presented study. The authors claim across the manuscript that Ssdp influences neurodevelopment, but all the data were conducted or collected after the development: from fully mature brains or adult flies. What is the evidence that links Ssdp expression to neurodevelopment? 

It is important for the authors to demonstrate that (i) brain structures and function at early developmental stages (e.g. larvae and pupae) are also affected in the condition of constant Ssdp overexpression and knockdown; (ii) and in a conditional experiment to exclusively manipulate Ssdp expressions in the adult brain there are no cellular and behavioral defects. These fairly experiments would significantly improve the rigor of the work.

2) Fig. 2C: What do Ssdp-positive neurons look like in Ssdp[2082-G4] flies? Are their structural components (axons, dendrites, soma) affected? These experiments would provide key insight at the cellular level into the neuro-developmental nature of this study.

3) Page 4 and Fig. 2E: I am curious about Ssdp upregulation in Ssdp[2082-G4]? Does an expression of UAS-SsdpRNAi driven by Ssdp[2082-G4] lead to a normalization of Ssdp levels?

4) The link between gene dosage and diseases is well-described in the literature. What are the consequences of an overexpression of Ssdp using Ssdp[2082-G4] or Elav-Gal4 on brain development and behaviors? 

5) It is unclear (Pages 6,7 and Figs. 3,4) how the authors explain the much larger neuropil, while the whole-brain morphology labeled by Elav or repo show no changes? Did author try another neuropil marker (e.g. NC82). 

6) Fig. 3C-F and Fig. 4D-G: Immunostaining Anti-repo or anti-Elav normally targets thousands of cells stacked on top of each other. How did the authors count Also, how did the authors count the neuron or glia cell bodies in the brains. A more detailed description of the counting method as well as the provision of all the dataset (e.g. Excel, tiff files) supporting this analysis are important to convince the readers.

7) What is the evidence for the specificity of the RNAi? 

In this manuscript the authors employ one RNAi transgene for Ssdp. RNA-interference is often conducted by expressing inverted repeats of cDNA fragments matching a gene of-interest using the Gal4/UAS system, as done in this study. However, this approach is notoriously prone to off-target effects. This is of particular concern when monitoring behavioral phenotypes, which are notoriously sensitive to genetic background.

- Has the SsdpRNAi line used in this manuscript (Fig. 4 and 8) been previously validated? Anyway, to what extent it reduce Ssdp expression? A simple experiment is to conduct a PCR experiment in the presence and absence of SsdpRNAi. 

- Can the authors observe effects on the brain (as in figure 4) as well as behaviors (as in Fig. 8) with additional SsdpRNAi transgenes?

8) On a related note, knockdown approach raises the question of what is the null phenotype? It may be the same palette of phenotypes but at a greater penetrance, or it may be a completely different phenotype. It is important for the authors to try other knockout tools (mutants) to determine the full LOF phenotype.

9) Figure 8: What is the behavioral consequences of Ssdp knockdown in Ssdp positive cells using Ssdp[2082-G4] driver? is there a rescue of phenotypes?

10) Page 7 and Fig. 5. Additional characterization of the influence of Ssdp on the head is needed. How does Ssdp overexpression influence the head? Is the eye, antenna or proboscis affected. Light microscope images of the Drosophila head should be a better way to show the connection between Ssdp and head development. These data would help readers understand the full scope of the findings.

11) Genotype of RNA sequencing: Why did the authors decide to only focus on the overexpression of Ssdp and not on its knockdown? Please clarify this section. 

13) Fig. 6: RNA sequencing was performed on RNA extracted from heads which include eye and cuticle. Since the requirement of SsdP (as can be noticed by its level of expression in the FlyAtlas database), verification of the list of main differentially expressed genes via the Quantitative Real-time RNA PCR extracted from the brain (not from the head) is necessary. This verification also concerns Fig.7C.

14) Page 13 and Fig. 8. Does a knockdown of Ssdp in sensory neurons affect the sensory perception and habituation phenotypes? Even if these experiments may seem outside the scope of the paper, discussing them at a minimal would be critical for the reader to understand the potential novelty of these results.

Minor

15) Information on the genotype of the lines used in this work are not clear. Maybe detail the full genotype in the legends. 

16) Fig.1. Alignment analysis: Further description of the method and tools/platforms in the text, or methods is needed.

17) How old are the adult flies used in this study for the molecular, cellular and behavioral assays?

Reviewer #2: The manuscript is focused on using Drosophila as a tool to study the role of genes located in regions affected by microdeletion/duplication in autism spectrum disorders (ASD) in the pathogenesis of these neurodevelopmental disorders. Changes in the dosage of these genes might be involved in these diseases but the pathogenic mechanisms are still unknown. In particular, the authors focused on SSBP3, Ssdp in the fruit fly ,and used both a knockdown and an overexpression strategy to assess the role of CNVs in neurodevelopment and autism-associated behaviour. The authors performed an interesting analysis of the consequences of altering Ssdp levels on different aspect of the animal biology, ranging from tissue development to gene expression alteration, to behavior. Overall, the conclusions are interesting and novel, nevertheless the neurodevelopment analysis is rather superficial and would require a more detailed analysis. In conclusion, I think this paper is a good candidate for Plos Biology but I would like to see the Major points addressed before considering it for publication.

Major points.

- The authors analyzed SSBP3 expression in human brain using already available scRNA seq data and conclude that it colocalizes with SLC17A7, CUX2, RORB and oligodendrocytes. Even though the heatmap shown in Figure 2A is quite convincing, the scatter plots (Figure 2B) are not. SSBP3 seems to be expressed at very low levels (if the color scale indicates level of expression, which is not mentioned in the text nor in the figure legend) in most of the domains where SLC17A7 is expressed and the same is true for RORB. Also, in the oligodendrocyte cluster there seem to be some small areas with higher expression of SSBP3 but the majority show very low expression levels. To confirm the colocalization in the different clusters of cells a better analysis is needed, maybe highlighting clearly which area the authors refer to when they speak of colocalization.

- To investigate the role of Ssdp during development the authors measured the neuropil area by staining the brain with anti-DLG antibody (Figure 3A,B). The threshold feature was used on maximum intensity projection images and the central area of the brain was selected (from methods). I assume the quantification of the neuropil has then been performed on these images. I think this is not a correct way of quantifying neuropil size in fact it does not take into account the flattening due to the mounting procedure. I do not think the authors used anything to normalize the quantification against this factor, except if all the brains quantified, both the controls and the Ssdp>GFP, were mounted on the same slide. The fact that this method is not the best can be also seen by the fact that the control brains in the overexpression and knowckdown experiments (Figure 3B and 4C) show significantly different neuropile areas, thing I do not think can be due to the genetic background. I think that a better evaluation of the neuropile area would come from the quantification of its volume, by considering all the stacks that compose the single image, since this would take into account the flattening. The analysis of neuronal area (Figure 3 C,D) is also very superficial and not properly performed. In the methods section it is indicated that the quantification of neuronal area has been performed in the same way as the neuropil area, nevertheless this is even less appropriate than in the case of anti-DLG antibody. Since Elav marks the nuclei of the neurons, performing the quantification of the neuronal area on a maximal projection is not correct, because it will not take correctly into account neurons present in different planes (in z) but with a similar x/y location. Moreover, if the Ssdp>GFP would affect neuronal size and neuronal number at the same time, i.e fewer bigger neurons, the area measurements would not be detecting such a defect. Finally, regarding the analysis of the Glia nuclei (Figure 3 E,F) also in this case the quantification is performed on a maximal intensity projection image. As clearly visible in Figure 3E w1118, there are areas where multiple nuclei are juxtaposed and overlapping, creating large positive areas, therefore I am wondering how this is taken into account when counting single nuclei. Did the authors used segmentation to be able to identify single nuclei in area such the one described? Otherwise, the quantifications are not accurate enough.

The same technical comments are valid for the experiments described in Figure 4.

In conclusion I think that for the paper to be considered for publication this part of the analysis (both the SsdpGal4 and the knockdown experiments) must be performed in better detail with appropriate controls: a) a quantification of the brain volume, rather than the brain area b) a quantification of the number of Elav positive cells and eventually size of cell bodies on the entire 3D of the brain c) a quantification of the glia cells in the 3D of the brain, based on all the single stack images, not on a maximal intensity projection.

- Regarding the expression of Ssdp in the brain, are the images showed in Figure 3G maximal intensity projection? Because if the authors want to investigate colocalization of the GFP with Elav or Repo it would be crucial to show single confocal planes or at least a maximal intensity projection of a number of confocal planes that covers only single layers of cell bodies.

- The authors analyzed whether the neuro-anatomical defects are associated with defective Wnt signaling and in Figure 5C they claim that although Armadillo and Wingless immunofluorescence were reduced, they were not statistically significant. Nevertheless, it seems that the pattern of expression of the two genes is slightly different in terms of number of positive stripes, could the authors comment? 

Minor points

- Relative to Figure 1: in the schematic representation of the genetic locations of the genes of interest is indicated the SSBP3-AS1 (antisense), but it is not mentioned in the text nor in the figure legend. If it's indicated because relevant to the study, it should be explained.

- The authors used SCOPE to determine the expression of Ssdp in relations to other genes: the colocalization with Repo and vGlut is clear but, since Ssdp expression seems to be much broader, I think it should be indicated as "partial" co-localization or as "Ssdp is expressed in Repo and vGlut positive cells.

- Relative to Figure 2: in the figure legend excitatory neurons are marked Exh and not Exc as in the scatter plot figure. Moreover, since the authors prove that the 2082Gal4, due to its nature, leads to increased mRNA levels of Ssdp, it would be important to mention if in the performed experiment the insertion is in homozygosis or in heterozygosis (either in the main text or in the figure legend).

- Can you indicate in the figure legend or in the test the meaning of the dotted lines that circle some areas of the brain in Figure 3E?

- In figure 5B I would suggest using a different layout for the dots in fact, since they are many and very close to each other, it becomes a bit crowded and confusing.

- In the figure legend of Figure 6I the p value and the number should be moved to the end of the sentence

- In Figure 7A the UAS-Mito-Red distribution seems very different between the control and the other conditions, for example much more spread and diffuse in the Mushroom Body area of the control, is this due to the changes in size and shape of the mitochondria?

- In the Figure 7 legend it is indicated wich control has been used for normalization of RT-PCR while in other is not indicated.

- In Figure 8J what is the meaning of the *? Please add an explanation in the figure legend.

- In Supplementary Figure 6B it is not clear which line is the W1118-Ssdp-RNAi, indicated in light grey below the graph.

- Figure 8H is not cited in the text

- Figure 9 is cited incorrectly in the text, average speed is 9B, not 9A, wall following behaviour is 9C, not 9B.

---

## [Decision Letter · Decision Letter 2]

31 May 2023

Dear Dr Farhan,

Thank you for your patience while we considered your revised manuscript "Ssdp influences neurodevelopment and autism-like behaviors in Drosophila melanogaster" for consideration as a Research Article at PLOS Biology. Your revised study has now been evaluated by the PLOS Biology editors, the Academic Editor and the original reviewers, who agree that the manuscript has been strengthened. However Reviewer 2 has some additional comments which we think will require further revisions to address. 

In light of the reviews, which you will find at the end of this email, we are pleased to offer you the opportunity to address the remaining points from the reviewers in a revision that we anticipate should not take you very long. We will then assess your revised manuscript and your response to the reviewers' comments with our Academic Editor aiming to avoid further rounds of peer-review, although might need to consult with the reviewers, depending on the nature of the revisions.

**IMPORTANT: As you address Reviewer 2's remaining comments, please also attend to the following editorial requests: 

1) TITLE: After some discussion within the team, we think the title should be modified slightly to make clearer the link between ssdp and human SSB3. If you agree, we suggest you change the title to "The ortholog of human single-stranded DNA-binding protein SSBP3 influences neurodevelopment and autism-like behaviors in Drosophila melanogaster"

2) FINANCIAL DISCLOSURES: I noticed in your financial disclosure that you report having received no specific funding for this work. Even if you did not have a grant specifically written to fund this project, we would require that you acknowledge the sources of money that funded this work, including salaries, etc. You should also acknowledge institutional funding or private grants as well, if relevant.

3) DATA: Thank you for depositing your RNA seq datasets on the GEO database - can you please provide me with a reviewer token so that I can access this data and make sure it complies with our data policies?

4) DATA: In addition to access to the RNA seq data, we ask that you provide the underlying data for each of your figures. You may be aware of the PLOS Data Policy, which requires that all data be made available without restriction: http://journals.plos.org/plosbiology/s/data-availability. For more information, please also see this editorial: http://dx.doi.org/10.1371/journal.pbio.1001797

a Supplementary files (e.g., excel). Please ensure that all data files are uploaded as 'Supporting Information' and are invariably referred to (in the manuscript, figure legends, and the Description field when uploading your files) using the following format verbatim: S1 Data, S2 Data, etc. Multiple panels of a single or even several figures can be included as multiple sheets in one excel file that is saved using exactly the following convention: S1_Data.xlsx (using an underscore).

b Deposition in a publicly available repository. Please also provide the accession code or a reviewer link so that we may view your data before publication. 

>>>Regardless of the method selected, please ensure that you provide the individual numerical values that underlie the summary data displayed in the following figure panels as they are essential for readers to assess your analysis and to reproduce it:

Fig 2E; Fig 4B,D,F,H; Fig 5A-F; Fig 6A-D; Fig 7I-J,L; Fig 8B-C; Fig 9A-E,G,I; Fig 10B-C,E,G;

Fig S2A; S7; S8; S9; S10; S11

>>>Please also ensure that figure legends in your manuscript include information on where the underlying data can be found, and ensure your supplemental data file/s has a legend.

>>>Please ensure that your Data Statement in the submission system accurately describes where your data can be found.

5) BLOT AND GEL REPORTING: We require the original, uncropped and minimally adjusted images supporting all blot and gel results reported in an article's figures or Supporting Information files. We will require these files before a manuscript can be accepted so please prepare and upload them now. Please carefully read our guidelines for how to prepare and upload this data: https://journals.plos.org/plosbiology/s/figures#loc-blot-and-gel-reporting-requirements

>>Please provide the original, uncropped blots accompanying Fig 6A. 

6) DATA NOT SHOWN: Please note that per journal policy, we do not allow the mention of "data not shown", "personal communication", "manuscript in preparation" or other references to data that is not publicly available or contained within this manuscript. Please either remove mention of these data or provide figures presenting the results and the data underlying the figure(s). I noticed one instance of this on page 13 “however downregulated genes were not enriched for any brain function related biological process GO terms (not shown)"

**IMPORTANT - SUBMITTING YOUR REVISION**

*Resubmission Checklist*

*Published Peer Review*

*PLOS Data Policy*

*Blot and Gel Data Policy*

Sincerely,

Lucas

Lucas Smith, Ph.D.

Senior Editor

PLOS Biology

lsmith@plos.org

REVIEWS:

Reviewer #1: The author answered the questions I raised, supported by additional figures or clarifications. These answers are compelling and I think the paper is now mature for publication.

Reviewer #2: The revised version of the manuscript addressed most if not all the points that have been raised by the reviewers and overall, the neurodevelopmental analysis has been substantially improved and strengthen by the new data and analysis. I have appreciated the way the authors answered my comments and the experimental strategies undertaken to solve the critical points and to improve the rigor of the study. I still have some minor comments regarding the revised version that I would like to see addressed:

1. I do appreciate the new volumetric analysis of the brains that takes into account all the stacks, and the new results are indeed more accurate. However, as I have highlighted in the previous rounds of comments regarding the Elav-immunostaining, I do not think the analysis performed as it is, gives any insight into the effect of Ssdp on brain volume. If Ssdp levels alteration affects at the same time neuronal number and size, no difference in volume would emerge and the defect would not be detected. This might be the case. In fact, in line 205-207 and 243-244, the authors use anti-Elav immunostainings to quantify brain volume without seeing any difference. I think that, if the authors would like to use Elav as a marker they should perform a quantification of cell numbers, as they did for Repo. 

2. I wonder why in the analysis of brain volume of Ssdp knockdown flies, the authors did not perform the anti-Bruchpilot analysis as in the overexpression flies (see lines 239-240 compared to 203-205).

3. Lines 282-284: I think it would be useful to cite again the figure panel the authors are referring to, after describing the differences in the number of stripes, maybe at the end of line 284.

4. In figure S7 the authors perform some RT-PCR on brains to confirm the data on mitochondrial fission/fusion genes obtained from heads extract (lines 1326-1327). Interestingly, Marf1 levels are not upregulated in the brains, while they are in the head extracts (Fig 8C) that also include eyes and cuticles. Does this suggest that the upregulation of Marf1 is dependent on these tissues? I would like the authors to comment on this.

5.In figure 9G the RNAi control (blue dots) seems to display an interaction time much closer to that of the Elav-RNAi flies, rather than to that of the other control (grey dots, Elav alone). Can the authors comment on this? In most of the other experiments it behaves very similar to the other control. Is the statistics calculated by pooling the two controls together and comparing them to the Elav-RNAi strain? What is the genotype of the two controls, are they outcrossed to a control strain? (i.e. Elav/+ and RNAi/+)

6. In figure S9 the authors show a partial rescue of behavioral defects upon knocking down Ssdp expression in Ssdp-overexpressing cells. Is there any statistical analysis performed between the Ssdp-Gal4 and Ssdp-Gal4>Ssdp-RNAi?

7. In the discussion, lines 640-641 the authors state that "Ssdp may plays an inhibitory or excitatory role in a tissue context-dependent manner", I think that "excitatory" is not the correct term for Ssdf function, maybe I would refer to "activatory".

8. In the discussion, lines 648-650 the authors state "Therefore, our findings suggest that Ssdp may play a critical role in glial cell function during neurodevelopment and maintenance of the nervous system". Since the evidence produced regards the number of glia cells, I think this statement should be tuned down.

9. In line 667-668 the authors state "Reduced Wnt/β-catenin signaling in Ssdp[2082-G4] brains may have resulted in larger brains and reduced gliogenesis". If the meaning of this sentence is that a reduction in Wnt signaling might represent the molecular mechanism underlying larger brain and reduced gliogenesis then the authors need to rephrase the concept

---

## [Editor Report · Decision Letter 3]

21 Jun 2023

Dear Dr Farhan,

Thank you for the submission of your revised Research Article "The ortholog of human single-stranded DNA-binding protein SSBP3 influences neurodevelopment and autism-like behaviors in Drosophila melanogaster" for publication in PLOS Biology. Your revised manuscript was assessed by the PLOS Biology editorial team and by our Academic Editor, and we are satisfied by the changes made in response to the reviewers and in response to our last editorial requests. Therefore, on behalf of my colleagues and the Academic Editor, Bassem A. Hassan, I am pleased to say that we can in principle accept your manuscript for publication, provided you address any remaining formatting and reporting issues. These will be detailed in an email you should receive within 2-3 business days from our colleagues in the journal operations team; no action is required from you until then. Please note that we will not be able to formally accept your manuscript and schedule it for publication until you have completed any requested changes.

**Important: as you address the formating requests to come, please also take a moment to update your data availability statement to reflect the newly added supplemental files containing the underlying data for your manuscript. In the relevant section of our online system please change your data availability statement to something like: "All RNA-seq files are available from the NCBI GEO database (accession number(s) GSE220311) and that all other relevant data are within the paper and its Supporting Information files"

PRESS

Sincerely, 

Lucas Smith, Ph.D.

Senior Editor

PLOS Biology

lsmith@plos.org